# Mechanism of NanR gene repression and allosteric induction of bacterial sialic acid metabolism

Christopher R. Horne [1,12], Hariprasad Venugopal[2,12], Santosh Panjikar [3,4], David M. Wood[1], Amy Henrickson [5], Emre Brookes[6], Rachel A. North[1], James M. Murphy [7,8], Rosmarie Friemann [9,10], Michael D. W. Griffin [11], Georg Ramm [2,4], Borries Demeler [5,6] & Renwick C. J. Dobson [1,11✉]

Bacteria respond to environmental changes by inducing transcription of some genes and repressing others. Sialic acids, which coat human cell surfaces, are a nutrient source for pathogenic and commensal bacteria. The *Escherichia coli* GntR-type transcriptional repressor, NanR, regulates sialic acid metabolism, but the mechanism is unclear. Here, we demonstrate that three NanR dimers bind a $(GGTATA)_3$-repeat operator cooperatively and with high affinity. Single-particle cryo-electron microscopy structures reveal the DNA-binding domain is reorganized to engage DNA, while three dimers assemble in close proximity across the $(GGTATA)_3$-repeat operator. Such an interaction allows cooperative protein-protein interactions between NanR dimers via their N-terminal extensions. The effector, *N*-acetylneuraminate, binds NanR and attenuates the NanR-DNA interaction. The crystal structure of NanR in complex with *N*-acetylneuraminate reveals a domain rearrangement upon *N*-acetylneuraminate binding to lock NanR in a conformation that weakens DNA binding. Our data provide a molecular basis for the regulation of bacterial sialic acid metabolism.

[1] Biomolecular Interaction Centre and School of Biological Sciences, University of Canterbury, Christchurch, New Zealand. [2] Clive and Vera Ramaciotti Centre for Cryo-Electron Microscopy, Monash University, Clayton, VIC, Australia. [3] Australian Synchrotron, ANSTO, Clayton, VIC, Australia. [4] Department of Biochemistry and Molecular Biology, Monash University, Clayton, VIC, Australia. [5] Department of Chemistry and Biochemistry, University of Lethbridge, Lethbridge, AB, Canada. [6] Department of Chemistry, University of Montana, Missoula, MT, USA. [7] Walter and Eliza Hall Institute of Medical Research, Parkville, VIC, Australia. [8] Department of Medical Biology, University of Melbourne, Parkville, VIC, Australia. [9] Department of Clinical Microbiology, Sahlgrenska University Hospital, Gothenburg, Sweden. [10] Centre for Antibiotic Resistance Research (CARe), University of Gothenburg, Gothenburg, Sweden. [11] Bio21 Molecular Science and Biotechnology Institute, Department of Biochemistry and Pharmacology, University of Melbourne, Parkville, VIC, Australia. [12] These authors contributed equally: Christopher R. Horne, Hariprasad Venugopal. ✉email: renwick.dobson@canterbury.ac.nz

Bacteria rapidly adapt to changes in nutrient availability. The physiological response to these changes is multi-layered, but a key element is gene regulation[1–3]. Genes that encode the appropriate metabolic machinery are induced, while those that are unnecessary are repressed. For example, the human gastro-intestinal tract is heavily populated with bacteria[4–7], but nutrient availability fluctuates[7–9] and glucose is often limiting[10–12]. Bacteria evolved the capacity to import and metabolize sialic acids, a diverse family of negatively charged, nine-carbon amino monosaccharides[13–15]. Sialic acids coat host cell surfaces and are abundant in the mucosal epithelia of the gastrointestinal tract, where they mediate a variety of physiological and pathological processes[16]. Sialic acids are also a source of carbon, nitrogen, and energy for pathogenic and commensal bacteria[17,18], and some species also incorporate these sugars onto their cell surface to evade the human innate immune response[14,19,20]. Bacterial sialic acid metabolism is largely confined to mammalian commensal or pathogenic bacteria and most of these species colonize sialic acid-rich areas[17,18], suggesting a link between sialic acid utilization and survival in the host.

In *Escherichia coli*, the Nan repressor (NanR) regulates the expression of proteins responsible for sialic acid uptake and metabolism[21] (Fig. 1a). As a transcriptional repressor, NanR binds to a DNA operator site containing three GGTATA repeats[21,22] located within the promoter region of target genes and downstream of the RNA polymerase-binding site thereby blocking transcription[18]. The GGTATA repeat operator is found in three operons (Fig. 1a–c), collectively referred to as the sialoregulon. This operon arrangement has been identified in *E. coli*, including Shiga toxin-producing strains, and *Shigella dysenteriae*[18,22].

*E. coli* NanR belongs to the GntR superfamily of transcriptional regulators, which comprise an N-terminal DNA-binding domain and a C-terminal effector-binding domain[23,24]. The DNA-binding domain has a highly conserved winged helix–turn–helix motif[25], while the C-terminal effector-binding domain can be divided into seven subfamilies based on their fold[24]. For GntR transcriptional regulators, the general mechanism of gene regulation is that the protein binds DNA through the N-terminal domain, thereby repressing gene transcription. To modulate repression, an effector molecule binds to the C-terminal domain and allosterically alters the conformation of the N-terminal domain[23,26–29], which in turn alters the affinity of the GntR regulator for DNA. *N*-Acetylneuraminate (Neu5Ac; Fig. 1d) is the most abundant sialic acid in humans[17,30] and the purported effector molecule for *E. coli* NanR[21,22]. However, the mechanism underpinning this allosteric effect is unknown, with no direct evidence confirming that Neu5Ac binds NanR or affects the NanR–DNA interaction.

Here we report that three NanR dimers cooperatively bind to the (GGTATA)$_3$-repeat operator and that cooperativity is mediated by a 32-residue N-terminal extension. The affinity of NanR for the (GGTATA)$_3$-repeat operator is weakened by the presence of Neu5Ac, suggesting this is the allosteric activator in vivo. The crystal structure of NanR (2.1 Å) in the presence of Neu5Ac reveals a conformation change that results in a new interface between the N-terminal and C-terminal domains, locking the protein into a conformation that would disrupt DNA binding in one monomer. To determine the mechanism of gene repression, single-particle cryo-electron microscopy (cryo-EM) structures of the NanR-dimer$_1$/DNA hetero-complex (3.9 Å) and the NanR-dimer$_3$/DNA hetero-complex (8.3 Å) were determined. When compared with the crystal structure, these models reveal a reorganization of the N-terminal domains upon DNA binding and highlight the proximity of each NanR dimer when bound to the (GGTATA)$_3$-repeat operator. Overall, these data uncover the molecular basis by which NanR represses the expression of genes that import and metabolize sialic acids in *E. coli*.

## Results

**NanR binds DNA cooperatively and with nanomolar affinity.** To determine the affinity of *E. coli* NanR for DNA, we performed electrophoretic mobility shift assays (EMSAs). Titrating NanR against FAM-5′-labeled double-stranded DNA containing the full (GGTATA)$_3$-repeat operator (Fig. 2a) resulted in the formation

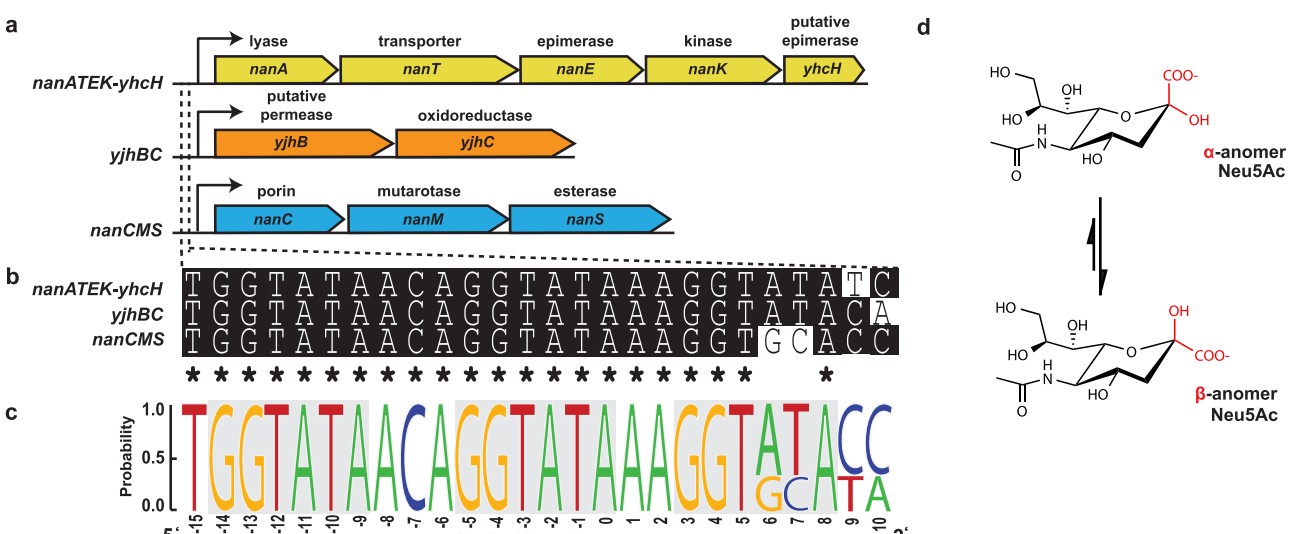

**Fig. 1 NanR binds a conserved operator site in three distinct operons. a** The sialoregulon consists of: *nanATEK-yhcH* (yellow), which is the core sialic acid catabolic pathway[18,21]; *yjhBC* (orange), which encodes proteins of unknown function that are hypothesized to process less common variants of sialic acid[18,85]; and *nanCMS* (blue), which is responsible for outer membrane transport and periplasmic processing[86-88]. **b** Sequence alignment of the operator sites present in each operon. Conserved nucleotides are marked with an asterisk. **c** Sequence logo highlights the conservation of the DNA bases within these operator sites (generated using WebLogo). The repeat sequence is shown in gray boxes. **d** The most common sialic acid, Neu5Ac, is shown in the chair conformation. The α-anomer and thermodynamically favorable β-anomer differ by the stereochemistry at the C2 position, highlighted in red.

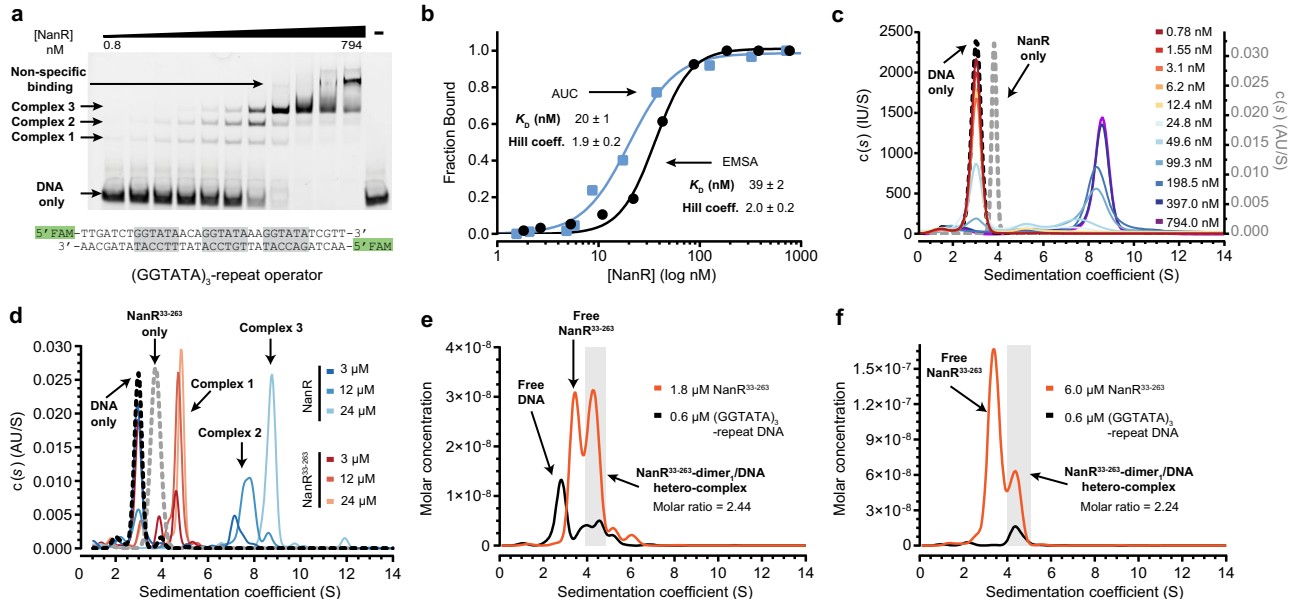

**Fig. 2 NanR binds DNA with positive cooperativity and nanomolar affinity. a** EMSA of NanR titrated against FAM-5′-labeled (GGTATA)$_3$-repeat DNA (10 nM, sequence shown below EMSA). Three concentration-dependent complexes are observed (complexes 1–3). Non-specific binding is observed at concentrations of NanR >200 nM. Data presented are representative of at least three independent experiments. **b** Binding isotherm data from the EMSA (black) and fluorescence-detected analytical ultracentrifugation (AUC) (blue) data best fitted the Hill model (AIC value of 99%), as opposed to a non-cooperative binding model (AIC value of 1%). The mean ± SD of the $K_D$ and Hill coefficient is shown. The independent experiment methods provide complementary data. **c** Continuous sedimentation coefficient [$c(s)$] distributions of NanR (gray dash), FAM-5′-labeled (GGTATA)$_3$-repeat DNA (black dash), and NanR (0.78–794 nM; rainbow) titrated against the FAM-5′-labeled (GGTATA)$_3$-repeat (80 nM). The binding isotherm data and fit are shown in **b**. The fitted parameters are shown in Supplementary Table 2. **d** Continuous sedimentation coefficient [$c(s)$] distributions of NanR and NanR$^{33-263}$ against the FAM-5′-labeled (GGTATA)$_3$-repeat (3 μM) (blue and orange traces, respectively) and NanR$^{33-263}$ alone (gray dash). Sedimentation was monitored via the absorbance of the FAM label at 495 nm. All experimental parameters are summarized in Supplementary Table 3. **e, f** Deconvoluted sedimentation coefficient distributions resulting from the titration of NanR$^{33-263}$ (orange) into (GGTATA)$_3$-repeat DNA (0.6 μM; black): 1.8 μM NanR$^{33-263}$ (**e**) and 6.0 μM NanR$^{33-263}$ (**f**). A shift in the sedimentation coefficient is observed with increasing NanR$^{33-263}$ concentration, consistent with hetero-complex formation. The molar ratio of the integrated peaks (shaded in gray) is that of a dimer. The presence of excess protein free of any co-migrating DNA in (**f**) indicates that hetero-complex formation has reached saturation. All plots are presented as $g(s)$ distributions with the molar concentration for each interacting partner (protein and DNA) plotted on the $y$-axis. Hydrodynamic parameters are in Supplementary Table 4.

of three hetero-complexes (Fig. 2a, labeled 1–3). NanR bound non-specifically at concentrations >200 nM, which was also observed in the poly-AT oligonucleotide control, where specific binding was abolished (Supplementary Fig. 1a). The ratio of bound to unbound DNA was determined by densitometry and was fitted to a Hill model with an apparent dissociation constant ($K_D$) of 39 ± 2 nM and a Hill coefficient ($n$) of 2.0 ± 0.2 (Fig. 2b, black line), evidence that NanR binding is cooperative.

Analytical ultracentrifugation studies corroborate this result. A DNA oligonucleotide containing the (GGTATA)$_3$-repeat operator sediments at 3.0 S, identifying the position of unbound DNA, while NanR sediments as a single peak at 3.70 S (Fig. 2c and Supplementary Table 1) corresponding to a dimeric oligomeric state (Supplementary Fig. 1b, c). Our binding assay monitors (by fluorescence) the sedimentation of the FAM-5′-labeled (GGTATA)$_3$-repeat operator sequence (80 nM) upon titration of NanR (0.78–794 nM). When the titration series was fit to a continuous sedimentation coefficient [$c(s)$] distribution, the peak corresponding to the free DNA decreases and three additional peaks develop (Fig. 2c and Supplementary Table 2), which mirrors the EMSA experiment (Fig. 2a). High affinity and cooperativity are again shown in a binding isotherm, determined by integrating across the NanR-bound DNA (3.5–12 S) and free DNA (2–3.5 S) peaks and fitted to the Hill model (Fig. 2b, blue line, $K_D =$ 20 ± 1 nM and $n = 1.9 ± 0.2$).

We next defined the number of GGTATA repeats required for NanR binding. NanR bound poorly to an oligonucleotide containing just one GGTATA repeat (Supplementary Fig. 1e), which was

similar to that of the poly-AT control (Supplementary Fig. 1a). However, while the binding affinity for the (GGTATA)$_2$-repeat oligonucleotide ($K_D = 25 ± 1$ nM, $n = 2.8 ± 0.3$) was similar to that of the (GGTATA)$_3$-repeat oligonucleotide, only two hetero-complexes were resolved (Supplementary Fig. 1f, g). Increasing the length of the spacers between the repeats by six nucleotides also attenuated NanR binding (Supplementary Fig. 1h). The requirement for two or more repeats with a defined spatial arrangement is consistent with cooperative binding, where elevated affinity arises from additional protein–protein interactions.

A comparative sequence analysis of homologous proteins (Supplementary Fig. 2) revealed that NanR has a 32-residue N-terminal extension within the DNA-binding domain. To test whether this extension plays a role in cooperativity, we generated a truncated NanR construct (NanR$^{33-263}$) and determined the effect of this truncation on (GGTATA)$_3$-repeat binding using analytical ultracentrifugation. Whereas for wild-type NanR, several species were evident at 7–9 S (Fig. 2d, blue traces, and Supplementary Table 3), for NanR$^{33-263}$ at the same concentrations only a single smaller species was evident at 4–5.5 S (Fig. 2d, red traces), demonstrating that although NanR$^{33-263}$ can bind DNA, it is unable to form the higher-order hetero-complexes. Together, these data implicate the N-terminal extension as a crucial determinant of cooperative assembly.

We next used analytical ultracentrifugation studies with multi-wavelength detection to demonstrate that just one NanR$^{33-263}$ dimer binds to the (GGTATA)$_3$-repeat operator. NanR$^{33-263}$ (1.8 or 6 μM)

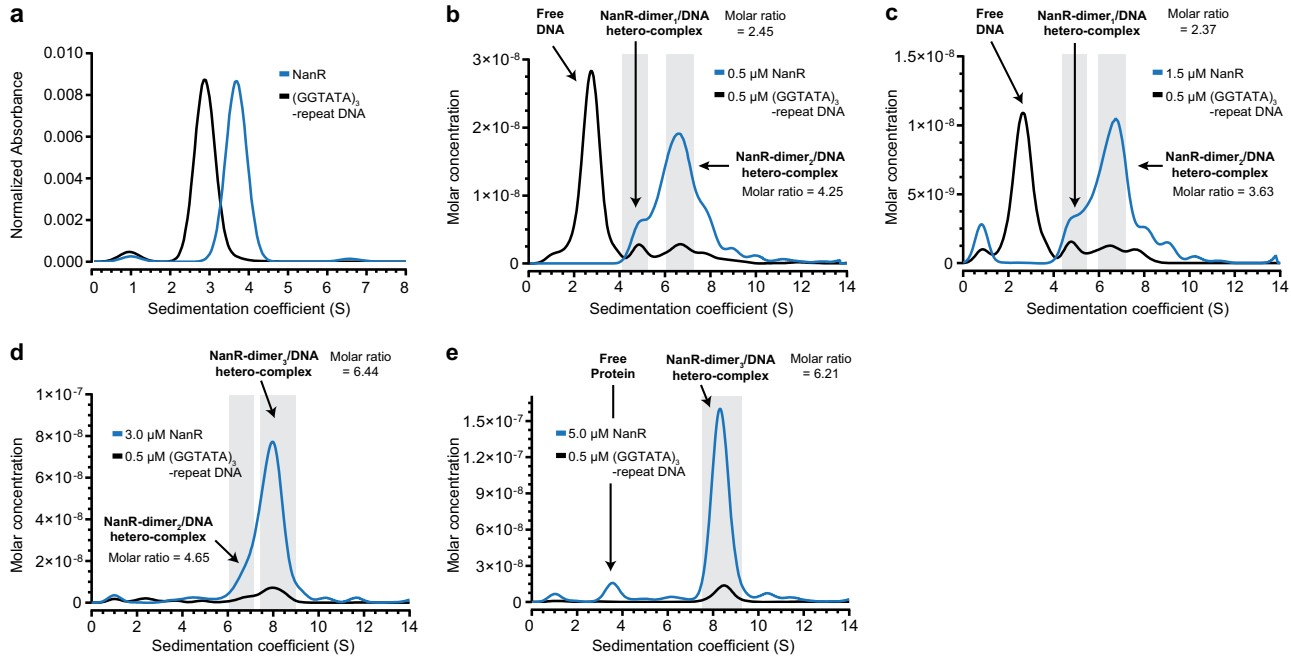

**Fig. 3 NanR dimers assemble on the (GGTATA)₃-repeat operator.** **a** Sedimentation coefficient distributions of the NanR (blue) and the (GGTATA)₃-repeat DNA operator (black) controls, measured individually at 280 and 260 nm, respectively. **b–e** Deconvoluted sedimentation coefficient distributions resulting from the titration of NanR into (GGTATA)₃-repeat DNA (0.5 μM): 0.5 μM NanR (**b**), 1.5 μM NanR (**c**), 3.0 μM NanR (**d**), and 5.0 μM NanR (**e**). A shift in the sedimentation coefficient is observed with increasing NanR concentration, consistent with hetero-complex formation. The molar ratio of the integrated peaks (shaded in gray) and the oligomeric state of each hetero-complex is shown. The presence of excess protein free of any co-migrating DNA in **e** indicates that hetero-complex formation has reached saturation. All plots are presented as g(s) distributions with the molar concentration for each interacting partner (protein and DNA) plotted on the y-axis. Hydrodynamic parameters are in Supplementary Table 5.

was mixed with (GGTATA)₃-repeat operator DNA (0.6 μM) and sedimentation velocity data were collected at multiple wavelengths (220–300 nm; Supplementary Fig. 3). Deconvoluting the multi-wavelength data for each interacting partner (i.e., NanR[33–263] and DNA) followed by analysis of the subsequent boundaries using a two-dimensional spectrum analysis (2DSA) method[31] generated molar concentration distributions for NanR[33–263] and DNA (Fig. 2e, f and hydrodynamic parameters in Supplementary Table 4). At a NanR[33–263] concentration of 1.8 μM (Fig. 2e), we found not only both free DNA (2.76 S) and free NanR[33–263] (3.42 S) but also co-migrating peaks at 4.0–5.0 S that were in agreement with complex 1 observed in Fig. 2d. Integrating the peak from 4.0 to 5.0 S (shaded area) gave a molar ratio of 2.44 NanR[33–263] monomers per DNA duplex. We next increased the concentration of NanR[33–263] to 6 μM (Fig. 2f) to saturate the DNA-binding sites, which resulted in the disappearance of the peak at 2.8 S, corresponding to free DNA, and an increase of the peak at 3.4 S corresponding to free NanR[33–263]. Integrating the co-migrating peaks (4–5 S, shaded area) gave a molar ratio of 2.24 NanR[33–263] monomers per DNA duplex. Together, these molar ratios suggest that a single NanR[33–263] dimer binds DNA, an assertion further supported by molar mass values that are consistent with the theoretical molar mass of a NanR-dimer₁/ DNA hetero-complex (Supplementary Table 4). Collectively, our data show that NanR binds the (GGTATA)₃-repeat operator with nanomolar affinity and that binding is cooperative, which is mediated by a unique N-terminal extension.

**Three NanR dimers bind the (GGTATA)₃-repeat operator.** Based on EMSA experiments, Kalivoda et al. proposed that a trimer, or a dimer followed by a monomer, initially binds the (GGTATA)₃-repeat operator to form complex 1[21,22], as seen in Fig. 2a. Inconsistent with these studies, our experiments show that

NanR is dimeric, with no evidence of a monomer in solution (Supplementary Fig. 1b). We sought to resolve this ongoing debate using analytical ultracentrifugation with multiwavelength detection to define the stoichiometry of the three hetero-complexes observed by EMSA (Fig. 2a). We titrated NanR (0.5–5.0 μM) against the (GGTATA)₃-repeat operator (0.5 μM) (Fig. 3). Compared with the distributions for separate protein and DNA controls (Fig. 3a), the deconvoluted distributions for the titration series demonstrated that the NanR and DNA peaks co-migrated (Fig. 3b–e), consistent with the formation of hetero-complexes. Integrating the co-migrating peaks between 4 and 5.25 S from the 0.5 and 1.5 μM data (Fig. 3b, c) gave molar ratios of 2.37:1 and 2.45:1, respectively, accordant with the formation of a NanR-dimer₁/DNA hetero-complex. Integrating the co-migrating peaks between 6 and 7.25 S gave molar ratios of 3.63:1 and 4.65:1 (Fig. 3b–d), consistent with a NanR-dimer₂/DNA hetero-complex. As the concentration of NanR was increased to 3 μM (Fig. 3d), a shift to a higher sedimentation coefficient (8.31 S) was observed. Integrating across the 7.5–9.25 S peak gave a 6.44:1 molar ratio, consistent with a NanR-dimer₃/DNA hetero-complex (Fig. 3d). At a NanR concentration of 5 μM (Fig. 3e), the molar ratio remained unchanged and we observed the presence of free protein, indicating that the system had reached saturation. The measured molar mass values for the peak at 8.3 S were 211 kDa (3 μM) and 205 kDa (5 μM), again consistent with the formation of a NanR-dimer₃/DNA hetero-complex (calculated molar mass is 198.5 kDa, Supplementary Table 5). Together these experiments show that discrete NanR dimers bind to the (GGTATA)₃-repeat operator to ultimately form a NanR-dimer₃/DNA hetero-complex.

**Neu5Ac attenuates the interaction between NanR and DNA.** There is no direct biophysical evidence confirming that Neu5Ac

binds to NanR to affect the NanR–DNA interaction. This led us to test whether Neu5Ac binds NanR, whether binding affects the oligomeric state of NanR, and what effect Neu5Ac binding might have on the affinity of NanR for DNA. First, to test whether Neu5Ac binds NanR, we performed differential scanning fluorimetric experiments. NanR exhibited a single unfolding transition in the first derivative plot (Supplementary Fig. 4a, black curve), with a transition melting temperature ($T_m^1$) of $52.0 \pm 0.1\,°C$ and an onset melting temperature ($T_{onset}$) of $48.8 \pm 0.1\,°C$. In contrast, the presence of Neu5Ac increased the $T_m^1$ to $54.0 \pm 0.1\,°C$ and the $T_{onset}$ to $50.0 \pm 0.8\,°C$, with a second transition melting temperature ($T_m^2$) evident at $68.1 \pm 0.1\,°C$ (Supplementary Fig. 4a, red curve). The second transition may reflect the increased thermal stability of the C-terminal effector-binding domain upon Neu5Ac binding. We next measured the dissociation constant ($K_D$) for Neu5Ac binding to NanR using isothermal titration calorimetry (Supplementary Fig. 4b), yielding a $K_D$ of $16\,\mu M$ (95% confidence interval 7–25 μM) and an $N$-value of 0.52, which is consistent with one Neu5Ac bound per NanR dimer.

To determine whether the presence of Neu5Ac disrupts the oligomer state of NanR, we performed analytical ultracentrifugation experiments using three NanR concentrations (3.3–30 μM). At each concentration, we observed a single species (3.65–3.71 S; Supplementary Fig. 4c), which is analogous with the sedimentation coefficient distribution observed in the absence of Neu5Ac (Supplementary Fig. 1b), supporting that NanR retains a dimer architecture in solution. There was no evidence of a monomer or larger oligomeric species, as suggested in previous crosslinking studies[22].

We next examined the effect of Neu5Ac binding on the NanR-DNA interaction by titrating NanR against the FAM-5′-labeled (GGTATA)₃-repeat operator sequence, in buffer supplemented with excess Neu5Ac (20 mM), and monitoring binding by fluorescence-detected analytical ultracentrifugation, using an analogous set-up as the experiment in the absence of Neu5Ac (Fig. 2c). In comparison to the data without Neu5Ac, there was a notable difference in the sedimentation coefficient distribution for the titration series when Neu5Ac was present in solution (Supplementary Fig. 4d and Supplementary Table 6), evidenced by an overall decrease in the signal for NanR–DNA hetero-complex formation (3.5–12 S) and an increase in the signal for free DNA (2-3.5 S), suggesting that Neu5Ac attenuates the NanR–DNA interaction. This attenuation was further illustrated in the binding isotherm, where the binding affinity for DNA decreased approximately 28-fold in the presence of Neu5Ac (Supplementary Fig. 4e, red line, $K_D = 578 \pm 26\,nM$ and $n = 2.0 \pm 0.6$), relative to the assay without Neu5Ac (Fig. 2b, blue line, and Supplementary Fig. 4e, black line, $K_D = 20 \pm 1\,nM$ and $n = 1.9 \pm 0.2$).

Taken together, these experiments demonstrate that one Neu5Ac molecule binds the NanR dimer with micromolar affinity and that binding does not alter the oligomeric state of NanR. Neu5Ac binding does, however, attenuate the affinity of NanR for the GGTATA recognition site, consistent with its proposed role in regulating sialic acid metabolism.

*NanR–Neu5Ac complex structure unravels the allosteric mechanism.* We solved the crystal structure of NanR in complex with Neu5Ac at 2.1 Å resolution to define how Neu5Ac binds NanR and, in turn, allosterically modulates the NanR–DNA interaction (data statistics in Supplementary Table 7). An X-ray fluorescence scan of NanR crystals suggested the presence of zinc (Supplementary Fig. 5a, b), which we exploited to solve the initial phases using single-wavelength anomalous diffraction. Inductively coupled plasma mass spectrometry of purified protein confirmed the presence of $Zn^{2+}$, with a 38-fold increase in $Zn^{2+}$ concentration (26.0 μg L$^{-1}$) in the protein solution compared with the buffer

$(0.7\,\mu g\,L^{-1})$. Zinc in complex with NanR was presumably carried through from expression, as it was not included in either the purification or crystallization conditions. Zinc is abundant in cells and 5–10% of proteins are predicted to bind $Zn^{2+}$ [32]. The ability to bind $Zn^{2+}$ groups NanR within a small and distinct family of GntR transcriptional regulators containing a metal-binding site[29,33]. Because the addition of $Zn^{2+}$ immediately precipitated purified protein, all studies reported here use NanR that was expressed from media and lysis buffer supplemented with $ZnCl_2$ (100 μM) to ensure maximum zinc occupancy.

Analogous to other GntR members, the NanR monomer has a two-domain architecture, comprising an N-terminal winged helix–turn–helix DNA-binding domain (Fig. 4a, α1–α3, β1–β2, green) linked to an α-helical C-terminal effector-binding domain (Fig. 4a, α4–α10, tan). The N-terminal DNA-binding domain has an antiparallel two-stranded β-sheet (Fig. 4a, inset blue) that defines the wing of the helix–turn–helix motif[23]. Helix α4 (Fig. 4a, purple) serves as a flexible linker connecting the N-terminal domain to the α-helical bundle of the C-terminal domain and is believed to play a role in the allosteric mechanism in GntR-family transcriptional regulators[23]. Helices α5–α10 arrange in an antiparallel bundle and play a role in both effector binding and dimerization (Fig. 4a, inset rainbow). These six α-helices in the C-terminal domain and helix α4 identify NanR as a member of the FadR subfamily[23,24,34].

The crystal structure reveals that NanR assembles into an asymmetric, domain-swapped dimeric architecture (Fig. 4b and Supplementary Movie 1), where the N-terminal domain is exchanged between monomers through the flexible α4-helix. The asymmetry of the dimer is driven by the presence of Neu5Ac and $Zn^{2+}$ in monomer A but not in monomer B. Neu5Ac, in the β-anomeric conformation, and $Zn^{2+}$ are bound together in a large polar cavity formed by the all-α-helical-bundle of the C-terminal domain. Neu5Ac is coordinated by a salt bridge with Arg128 (Fig. 4c), and Asn165, Asp172, His176, Arg203, His214, Asn215, Ser218, Gln221, and His244 through direct or water-mediated hydrogen bonds, as well as Phe168, Ile200, and Leu245 through hydrophobic interactions (Supplementary Fig. 5g). The $Zn^{2+}$ ion is coordinated in an octahedral geometry, interacting with the carboxyl and hydroxyl moieties of Neu5Ac and the sidechains of Asp172, His176, His222, and His244 (Fig. 4c and Supplementary Fig. 5g) with bond lengths ranging from 2.0 to 2.2 Å.

Capturing the Neu5Ac-bound conformation in one monomer (Chain A), while the opposing monomer (Chain B) is Neu5Ac-free (Fig. 4d), provides insights into how Neu5Ac attenuates DNA binding. Superimposition of the C-terminal effector-binding domains (root-mean-square deviation (RMSD) = 1.89 Å) demonstrated that the helices close in and around Neu5Ac-$Zn^{2+}$. The largest change was a rearrangement in the α8–α9 loop, allowing Arg203 to interact with the carboxyl moiety of Neu5Ac (Fig. 4d). In addition, Arg128 on the α5-helix also binds the carboxyl of Neu5Ac, which pulls the α5-helix away from the flexible α4-helix linking the effector-binding and DNA-binding domains. Together, these movements disrupt hydrophobic interactions with the α4-helix, changing the position of the C-terminal domain relative to the α4-helix. Superimposition of the monomers gave a RMSD of 4.55 Å over 207 equivalent Cα atoms (Fig. 4e) and showed that the binding of Neu5Ac compresses the NanR monomer around the α4-helix by 18.9 and 9.4 Å at the N- and C-terminal, respectively (Fig. 4f). We observed that the DNA-binding domain moved 22 Å and over 10.5° (Fig. 4e), placing it close to the C-terminal domain of the opposing monomer where it formed an extensive new interface locking the N-terminal domain in a closed conformation. In contrast, the apo conformation of NanR has fewer interactions, primarily via a salt bridge between Arg47 (N-terminal domain) and Asp197′ (C-terminal domain), while Arg170′ interacts with

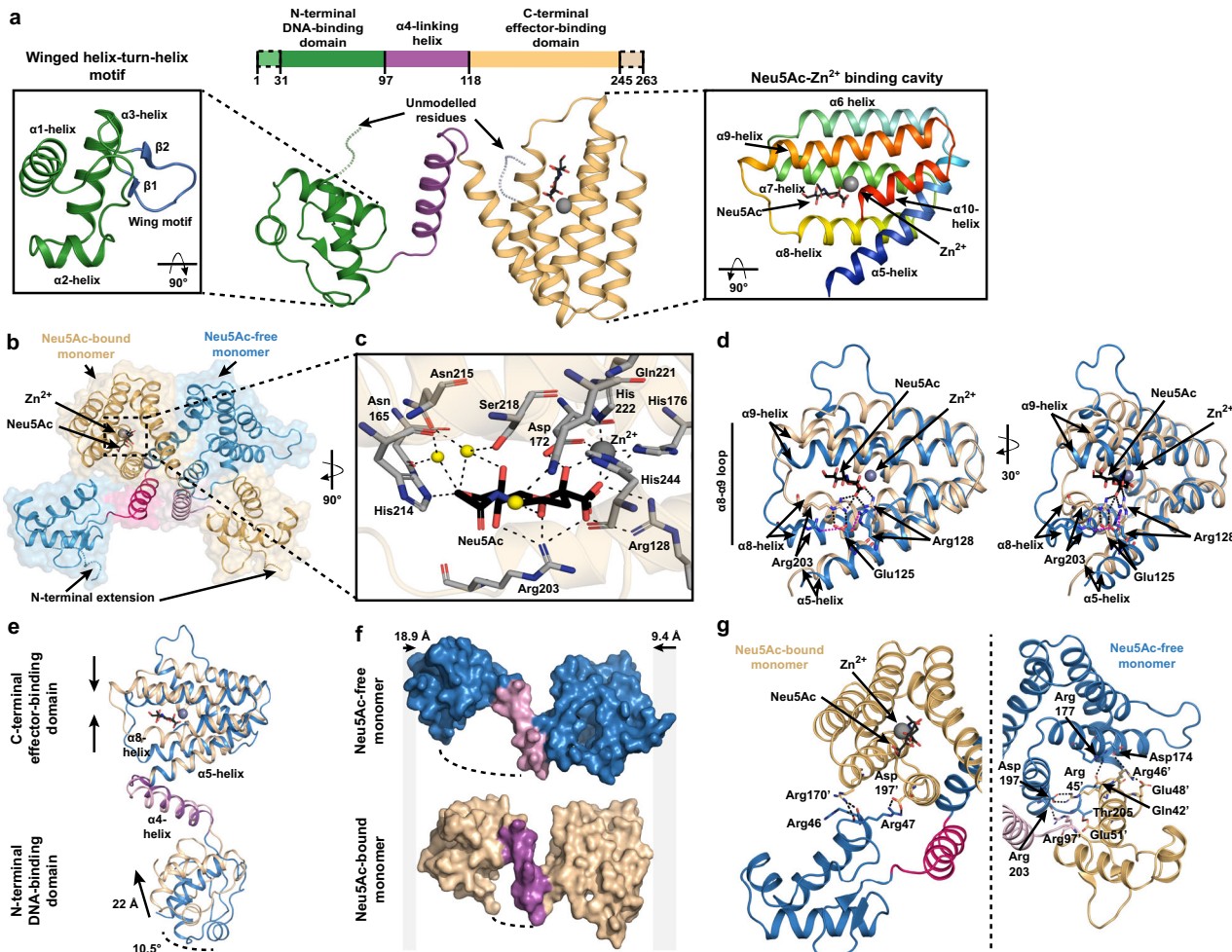

**Fig. 4 Crystal structure of NanR in complex with Neu5Ac and $Zn^{2+}$. a** The *E. coli* NanR monomer has two domains—an N-terminal DNA-binding domain (green) and C-terminal effector-binding domain (beige). The DNA-binding domain contains a highly conserved winged helix–turn–helix motif (left panel) where the wing is defined by an antiparallel two-stranded β-sheet (blue). The C-terminal domain is arranged into an antiparallel, all-α-helical bundle (right inset, rainbow). Helix α4 (purple) is a flexible linker connecting the two domains. **b** Cartoon/surface representation of the asymmetric domain-swapped dimeric structure formed by an exchange between monomers via the α4-helix (pink). The Neu5Ac-bound and Neu5Ac-free monomers are shown in beige and blue, respectively. **c** The effector binding site is located within a large polar cavity of the C-terminal domain. The direct or water-mediated hydrogen-bonding residues (gray sticks) that coordinate Neu5Ac in its β-anomeric form and hold $Zn^{2+}$ in an octahedral geometry are indicated, while water molecules are depicted as yellow spheres. **d** An overlay of the Neu5Ac-bound C-terminal domain (beige) and Neu5Ac-free C-terminal domain (blue) illustrates the effector-induced conformational changes. **e** An overlay of the Neu5Ac-bound monomer (beige) and Neu5Ac-free monomer (blue) further illustrates effector-induced conformational changes. **f** Surface depiction of Neu5Ac-bound (beige) and Neu5Ac-free (blue) monomers shows that the binding of Neu5Ac compresses the monomer around the α4-helix (purple) relative to the Neu5Ac-free monomer by 28.3 Å. **g** Cartoon representation of the interface between the Neu5Ac-bound monomer (beige) and the Neu5Ac-free monomer (blue), facilitated by salt-bridge interactions.

the main chain of Arg47 and Arg46 (Fig. 4g). Small-angle X-ray scattering experiments comparing NanR alone and NanR in the presence of Neu5Ac show a decreased $R_g$ value (32.5–31.4 Å) (Supplementary Fig. 6a and Supplementary Table 8), supporting the observation that Neu5Ac compacts the protein in the crystal structure. Further, the Neu5Ac-free scattering data best fit the extended symmetrical apo model ($X^2$ value of 3.7 using CRYSOL; Supplementary Fig. 6a), while the scattering data in the presence of Neu5Ac best fit the compact asymmetric crystal structure ($X^2$ value of 6.6 using CRYSOL; Supplementary Fig. 6a), rather than a symmetrical Neu5Ac-bound model ($X^2$ value of 11.3 using CRYSOL; Supplementary Fig. 6a). This suggests that only one Neu5Ac molecule has bound NanR, which is consistent with the stoichiometry ($N$-value of 0.52) obtained from our isothermal titration calorimetry experiments (Supplementary Fig. 4b).

**NanR-dimer₁/DNA hetero-complex reveals the mechanism of DNA binding.** To define how NanR binds the DNA operator, we determined the single-particle cryo-EM structure of the 70.5 kDa NanR-dimer₁/DNA hetero-complex at 3.9 Å resolution (workflow in Supplementary Fig. 7 and data statistics in Supplementary Table 9). NanR binds DNA in an asymmetric pose relative to the DNA helix (Fig. 5a and Supplementary Movie 2). This binding mode is supported by small-angle X-ray scattering data for the hetero-complex, which gave a $X^2$ value of 2.3 when compared with the theoretical scatter of the cryo-EM structure (Supplementary Fig. 6b). We used the (GGTATA)₂-repeat oligonucleotide to solve this structure (Fig. 5b) as NanR bound poorly to the oligonucleotide with only one GGTATA repeat (Supplementary Fig. 1e). The C-terminal domain closely matched that found in the NanR crystal structure (Fig. 5c, RMSD = 2.1 Å), showing that

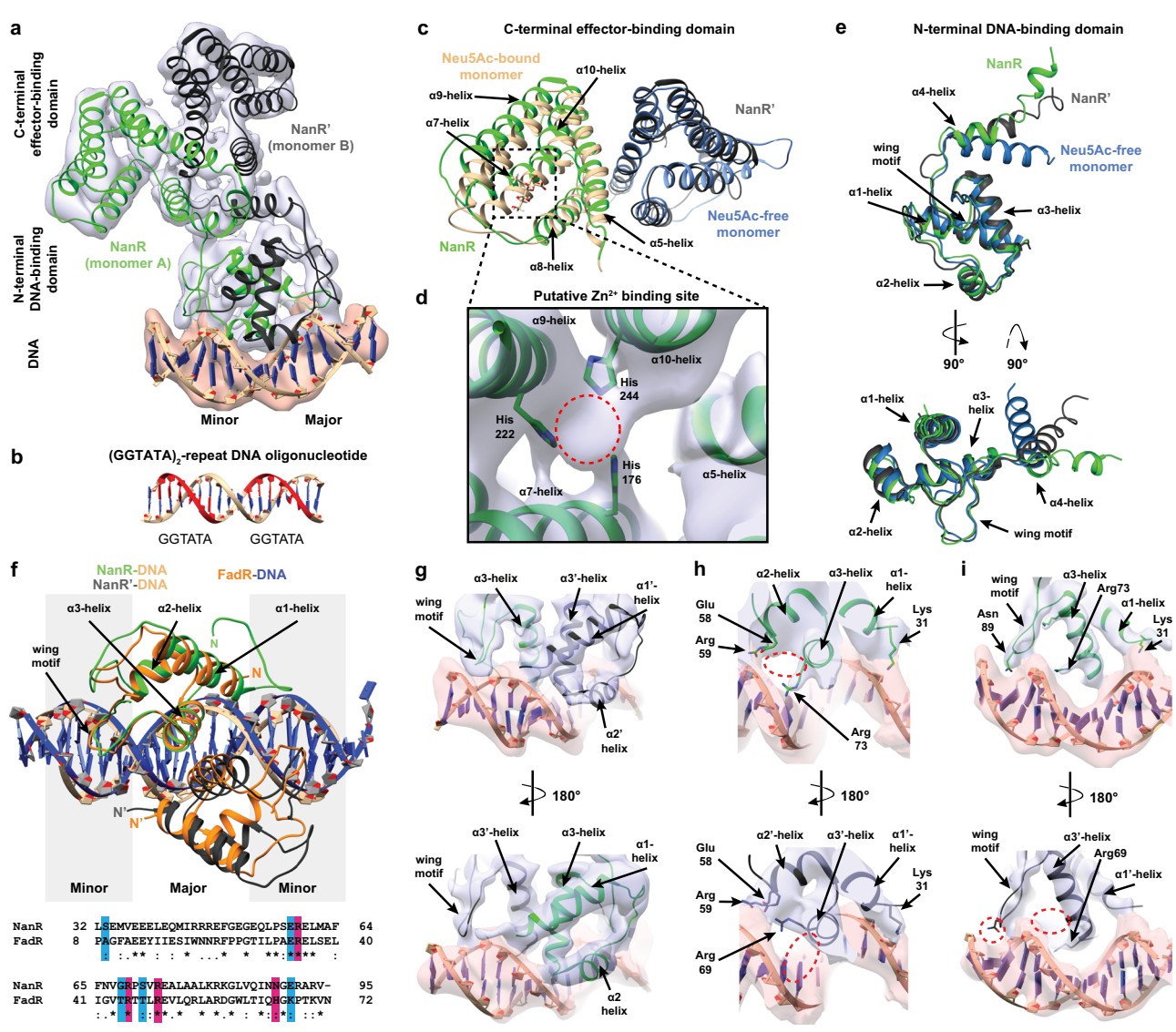

**Fig. 5 Cryo-EM structure of the NanR-dimer₁/(GGTATA)₂-repeat DNA complex. a** A 3.9 Å resolution cryo-EM reconstruction of the *E. coli* NanR-dimer₁/ DNA hetero-complex. Reconstruction is shown as transparent isosurfaces fitted with the cartoon representation of the hetero-complex structure. **b** The (GGTATA)₂-repeat DNA oligonucleotide used to solve this dataset, where each repeat is highlighted red. **c** Overlay of the C-terminal effector-binding domain from the crystal structure (beige and blue) and the cryo-EM structure (green and black) closely match each other (RMSD = 2.1 Å). **d** Density in the cryo-EM reconstruction that is hypothesized to be Zn²⁺ (red circle). Coordinating histidine residues are shown as sticks. **e** Overlay of the N-terminal DNA-binding domain and the flexible α4-linking-helices from the crystal structure (blue) and DNA-bound cryo-EM model (green and black). Note the change in direction of the α4-helix between structures. **f** Overlay of the interaction between the N-terminal DNA-binding domain and DNA for NanR (green, black, and beige) and FadR (orange and blue) [PDB ID: 1HW2]. A sequence alignment of the N-terminal domain for each structure shows that many of the DNA-binding residues in FadR are conserved in NanR. An asterisk indicates fully conserved residues, a colon indicates conservation between residues of strongly similar properties, and a period indicates conservation between residues of weakly similar properties. Residues proposed to interact with the phosphate backbone are highlighted blue, while residues proposed to interact with the DNA base pairs are highlighted in pink. **g–i** Close up of the N-terminal domain to highlight the difference in DNA binding between monomer A (green) and B (black). Sidechains of putative DNA-binding residues that could be resolved are shown (e.g., Arg73 and Asn89). A red circle highlights where a sidechain could not be resolved in the opposing monomer.

DNA binding does not markedly alter C-terminal domain architecture. Density for the α10-helix extended further than the crystal structure, allowing additional residues of the C-terminus in both monomers to be modeled. We also observed density between the α7- and α9-helices (evident across different thresholds) corresponding to the zinc-binding site in the crystal structure, including density for the histidine sidechains (Fig. 5d). We note that the α3-helix (~4.0 Å) and the α5-helix (~3.7 Å) at the dimer interface present the highest local resolution within the model (Supplementary Fig. 8c, d).

Analogous to the overlay of the C-terminal domain, the N-terminus closely matched that found in the crystal structure (Fig. 5e), showing that DNA binding does not substantively alter N-terminal domain architecture. However, the direction, length, and position of the α4-linking helices are altered when compared to the crystal structure (Fig. 5e). In the crystal structure, the α4-linking helices are compact and cross to form the domain-swapped monomers (Fig. 4b and Fig. 5e, blue). In contrast, when bound to DNA in the cryo-EM structure, we observed that the α4-linking helices are oriented in a different direction and adopt a

more extended conformation (Fig. 5e, green and black). This overlay also showed a difference in morphology between each monomer of the DNA-bound structure, supporting an asymmetric NanR–DNA interaction. Nevertheless, the conformational change between the crystal and cryo-EM structures results in a repositioning of the N-terminal domains as they engage DNA.

Reconstruction of the DNA oligonucleotide in the cryo-EM density for this dataset is unambiguously guided by the major and minor grooves of DNA (Fig. 5a). The α2- and α3-helices of each N-terminal DNA-binding domain make contact at the major groove, whereas the α1-helices interact with the DNA phosphate backbone (Fig. 5f). This binding mode is analogous to the GntR-type transcriptional regulator FadR (PDB ID: 1HW2)[34]. Superimposition of the equivalent FadR-DNA structure gives an RMSD of 0.984 Å and sequence alignment shows many of the DNA-binding residues in FadR are conserved in NanR (Fig. 5f). For FadR, the α3-helices also bind within the major groove, while the wing motif interacts with the minor groove, analogous to NanR in our cryo-EM model. Based on our cryo-EM structure of the NanR-dimer$_1$/DNA hetero-complex, the above sequence comparison with FadR, a mutational analysis performed by Kalivoda et al.[22], and sidechain chemistry (i.e. positive charge), we have identified nine putative DNA-binding residues. Ser33 in the α1-helix, Glu58 in the α2-helix, Gly68 and Ser71 in the α3-helix, and Glu91 in the wing motif are likely to form an interaction with the phosphate backbone of DNA, while Arg59 in the α2-helix, Arg69 and Arg73 in the α3-helix, and Asn89 in the wing motif

likely make sequence-specific contacts with the DNA bases within the operator sequence (Fig. 5f). There was a clear difference in local resolution between the two N-terminal domains. Comparatively, monomer A is better resolved, particularly in the wing motif and the α3-helix (Fig. 5g–i, top panel), which allowed several of the putative DNA-binding sidechains, such as Arg73, to be assigned in the model. This suggests a difference in binding affinity between the N-terminal DNA-binding domains to the non-equivalent DNA-binding sites. Despite the assignment of these putative DNA-binding residues based on these inferences, it is important to note that the resolution of the overall DNA-binding region (~5 Å; Supplementary Fig. 8c) is insufficient to resolve specific DNA base pair contacts with the (GGTATA)$_2$-repeat oligonucleotide. That said, this asymmetry in the DNA-binding pose suggests that there is a difference in affinity between the DNA-binding domains and the non-equivalent DNA-binding sites.

### α4-helices play a fundamental role in the allosteric mechanism of NanR.

Together, our crystallography and cryo-EM experiments allow us to define the molecular choreography that occurs when NanR binds DNA to repress gene expression or Neu5Ac to induce gene expression. The apo-NanR model, generated from the crystal structure, has a dimeric conformation, where the N-terminal DNA-binding domains are flexible (Fig. 6a, structure in blue), evidenced by the very few connections between the N- and

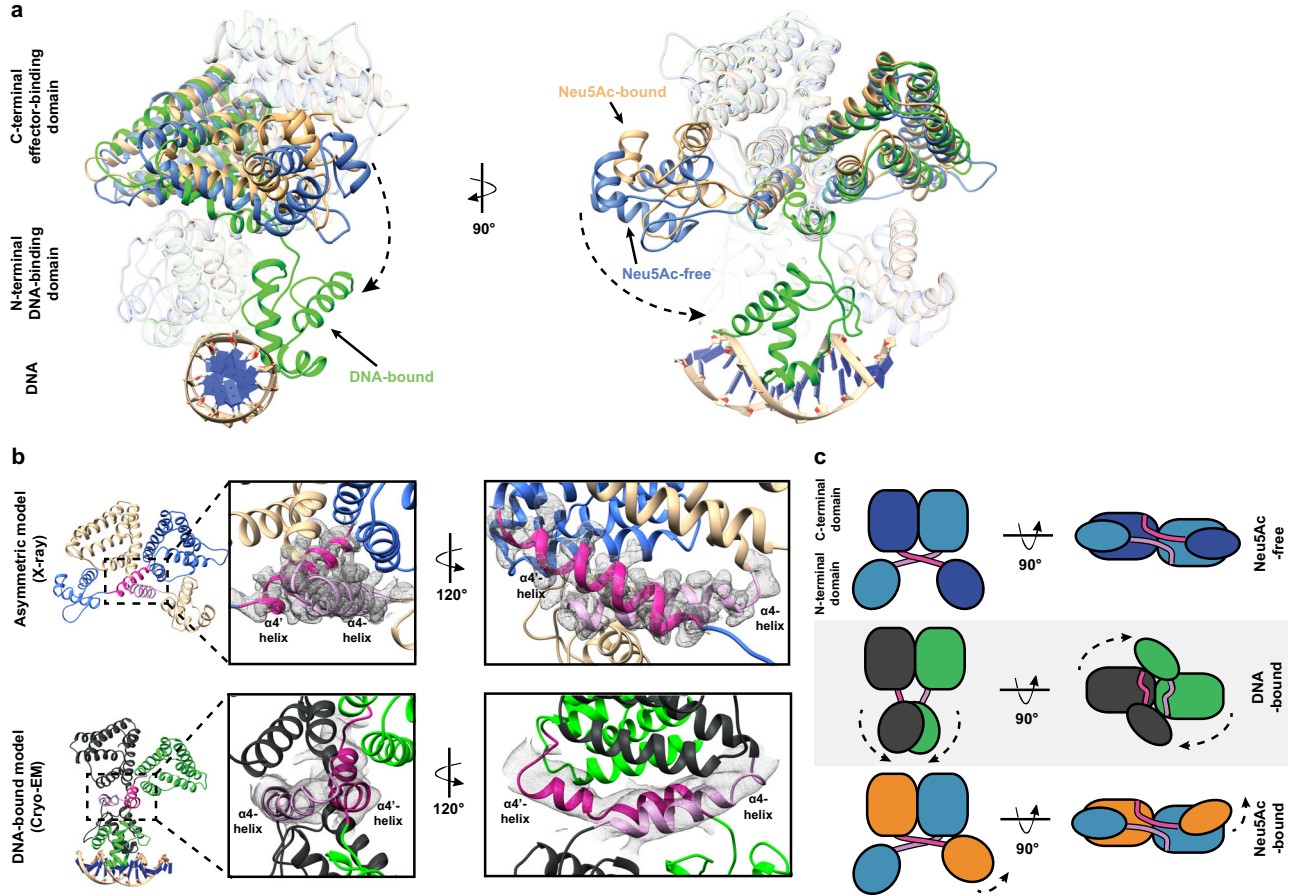

**Fig. 6 NanR undergoes large conformational changes on DNA binding. a** The three structural states of *E. coli* NanR (Neu5Ac-free (cyan), Neu5Ac-bound (beige), and DNA-bound (green)) are superimposed to highlight the conformational changes that occur as part of the allosteric mechanism. Notably, the N-terminal DNA-binding domain is reoriented via the flexible α4-helices (dash arrows). **b** The electron density between models shows that the α4-helices (light and dark pink) mediate a domain-swapped interface in the crystal structure (upper panel); however, these are no longer domain swapped when bound to DNA (lower panel). **c** Schematic to illustrate the structural changes the domains undergo between each state.

C-terminal domains (Fig. 4f, g). An overlay of the apo-NanR model with the DNA-bound cryo-EM structure revealed the most prominent change induced by DNA binding is a large reorganization of the N-terminal domains (Fig. 6a, structure in green) as they swing down to engage the major and minor grooves of the DNA—a conformational change that is facilitated by the α4-linking helix (Supplementary Movie 3). In the crystal structure, the α4-linking helices cross to form the domain-swapped monomers (Fig. 6b, upper panel, and Fig. 6c). In contrast, when bound to DNA in the cryo-EM structure, these helices are no longer domain-swapped (Fig. 6b, lower panel, and Fig. 6c). This would require that the N-terminal domains untwist before or upon DNA binding, which is plausible given their flexibility evident in the apo-NanR structure. This conformational change of the α4-linking helices can unambiguously be observed in the density maps between the crystal and cryo-EM structures (Fig. 6b). Neu5Ac binding promotes an opposite conformation (Fig. 6a, beige structure), whereby the N-terminal domain of one monomer moves closer to the C-terminal domain of the opposing monomer, allowing new interactions to be formed (Figs. 4g and 6c). This would lock the Neu5Ac-bound structure in a conformation that would be unfavorable for DNA binding, reducing the affinity for the NanR–DNA interaction. Taken together, these structural studies illustrate that the α4-linking

helix plays a fundamental role in the mechanism of NanR gene repression and allosteric induction.

**Three NanR dimers closely assemble across the (GGTATA)₃-repeat DNA.** To define how NanR engages the (GGTATA)₃-repeat operator, we determined an 8.3 Å resolution cryo-EM structure of the 198.5 kDa NanR-dimer₃/DNA hetero-complex (workflow in Supplementary Fig. 9 and data statistics in Supplementary Table 9). Initial two-dimensional classifications revealed two distinct populations: population 1 that had three NanR dimers bound to DNA, consistent with a NanR-dimer₃/DNA hetero-complex (Supplementary Fig. 9c); and population 2 that had a mixture of one or two NanR dimers bound to DNA (Supplementary Fig. 9d). However, due to the limited particle numbers and comparatively weaker signal of the resultant class averages compared to population 1, three-dimensional (3D) reconstruction was not suitable for the population 2 dataset.

3D reconstruction of the NanR-dimer₃/DNA hetero-complex within population 1 revealed sufficient density for the (GGTATA)₃-repeat operator and three NanR dimers (Fig. 7a, b), confirming the stoichiometry from our analytical ultracentrifugation experiments (Fig. 3d, e). We could unambiguously rigid body fit two NanR dimers, solved in our initial cryo-EM experiments, at either end of

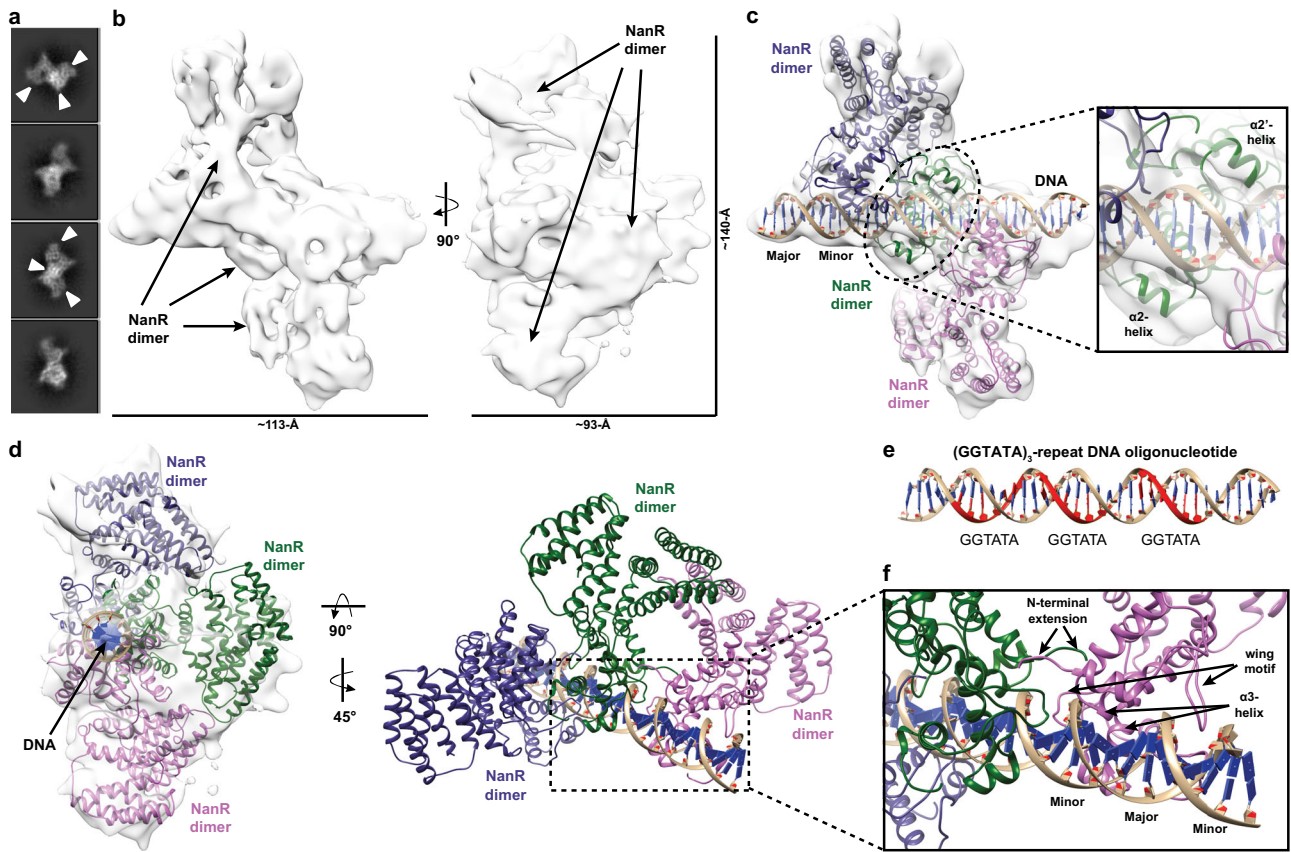

**Fig. 7 Cryo-EM structure of the NanR-dimer₃/DNA hetero-complex. a** 2D class averages showing three possible NanR dimers bound to DNA in projection (white arrow). **b** Density map for the NanR-dimer₃/DNA hetero-complex. **c** Three NanR dimers and a DNA duplex were fitted using a rigid body method. Two NanR dimers are clearly visible within the density (gray isosurface) at each end of the DNA, while the middle dimer is less resolved as a result of orientation bias. The DNA flanking α2-helices provide fit confidence for the middle NanR dimer (green, inset). **d** Each NanR dimer is offset by approximately one half turn of the DNA helix. **e** Each GGTATA repeat of the DNA operator (in red) is separated by less than half a turn of the DNA helix. **f** Zoomed view of the NanR–DNA interface highlighting how each NanR bound to the (GGTATA)₃-repeat operator maintains an analogous binding mode to the NanR-dimer₁/DNA hetero-complex (Fig. 5), where the α3-helix binds in the major groove and the wing motif accommodates the minor groove of DNA. The N-terminal domains are hypothesized to interact with each other, given their proximity, through protein–protein interactions in the higher-order hetero-complex.

the DNA (Fig. 7c). Analogous to the NanR-dimer$_1$/DNA hetero-complex dataset, reconstruction of the (GGTATA)$_3$-repeat oligo-nucleotide in the cryo-EM density was unambiguously guided by the grooves of DNA. However, density for the central NanR dimer had a lower signal-to-noise ratio and was consequently less resolved (Fig. 7b and Supplementary Fig. 9c). Despite this, the middle NanR dimer could be confidently placed as densities corresponding to the flanking α2- and α2′-helices of the N-terminal domain were observed when looking from the bottom of the DNA toward the dimer interface (Fig. 7c, inset).

Analogous to the DNA-bound structure above (Fig. 5a), we observe that each NanR dimer sits in an asymmetric pose relative to the DNA, approximately a half turn away from each other (Fig. 7d and Supplementary Movie 4), which aligns with the location of each GGTATA repeat within the operator sequence (Fig. 7e). Furthermore, we identified that the α3-helix made primary contact with the major groove, while the wing motif interacted with the minor groove of DNA across all three NanR dimers (Fig. 7f). Although the resolution of this dataset (Supplementary Fig. 10b) does not allow us to accurately locate the N-terminal extensions or confidently define their role in the assembly process, we note that they would be well placed to form protein–protein interactions with the adjacent NanR dimers to stabilize the complex (Fig. 7f).

## Discussion

Here, we characterize in molecular detail the mechanism by which NanR, a GntR-type gene regulator, represses the expression of genes that import and metabolize sialic acids in *E. coli*. Our biophysical studies demonstrate that three dimers of NanR sequentially bind the (GGTATA)$_3$-repeat operator with low nanomolar affinity, which is unusual for a GntR-type transcriptional regulator. This result differs from previous studies[22], where the first stable complex was proposed to be trimeric. Like most members of the GntR superfamily, which function as dimers[35,36], our biophysical and crystallographic studies demonstrate that NanR forms an obligate dimer, with no evidence for the trimeric or monomeric states needed to form an initial trimeric complex with DNA.

We demonstrate that the high affinity of NanR for the (GGTATA)$_3$-repeat operator sequence is driven by cooperativity. Interaction studies by EMSA and analytical ultracentrifugation both give a binding isotherm that fits the Hill model with a Hill coefficient of ~2. Moreover, high affinity binding requires two or more repeats in a close spatial arrangement, evidenced by the lack of binding to either a single GGTATA repeat sequence or increasing the length of the spacers between the repeats, suggesting that the dimers interact in some way. We defined the region of NanR that is responsible for cooperative binding to within the 32-residue N-terminal extension of the DNA-binding domain, as removal of this extension abolished assembly of the higher-order oligomers. Interestingly, we note that this N-terminal extension of NanR is significantly larger than those found in closely related GntR-type regulators (Supplementary Fig. 2), suggesting that the mechanism adopted by NanR to maintain tight, coordinated control of the sialoregulon differs from other modes of homotropic cooperative binding for GntR-type regulators reported to date. These previously described binding modes are typically driven by protein–protein interactions between neighboring protomers[37] and include the GntR-type regulators CitO[38] and PhnF[39], which involve two binding sites, and the *lac*[40] and *ara*[41–44] repression systems, which involve DNA looping.

Although the precise identity of the allosteric modulator of NanR is unclear, in vivo studies suggest that Neu5Ac induces the

sialic acid catabolic pathway[18,21,22]. Moreover, Kalivoda et al. used crosslinking studies to show that Neu5Ac binding disrupts the oligomeric state of NanR, abolishing DNA binding and inducing gene expression[21,22]. In agreement with this model, our binding studies demonstrate that NanR binds Neu5Ac with micromolar affinity and a stoichiometry of one Neu5Ac molecule per NanR dimer, which is consistent with our crystallographic and solution studies. However, in contrast with the Kalivoda et al. model, our biophysical studies reveal that NanR retains its dimeric structure with or without Neu5Ac present. By measuring the protein–DNA interaction, using nanomolar concentrations of NanR above and below the reported $K_D$, and saturating concentrations of Neu5Ac, we observed that the presence of Neu5Ac attenuates DNA binding 28-fold. This large change in DNA-binding affinity, in concert with the Neu5Ac-induced conformational change identified in our crystal structure, demonstrates that the mechanism of induction adopted by NanR is consistent with the classic allosteric mechanism employed by other members of the GntR family. These data support a model in which effector binding induces dissociation of the repressor from the DNA operator[23].

Our cryo-EM structure of a NanR dimer bound to a (GGTATA)$_2$-repeat sequence shows that the repressor binds DNA in an asymmetric pose. Interestingly, the N-terminal DNA-binding domains engage DNA in a manner that is analogous to FadR, a closely related GntR-type regulator. However, unlike FadR, where each N-terminal domain binds a palindromic DNA sequence symmetrically, NanR binds a repeat sequence with one N-terminal domain of the dimer engaging the consensus (GGTATA) sequence and the other N-terminal domain binding an adjacent non-consensus DNA sequence. The local resolution of each N-terminal domain was considerably different in the NanR-dimer$_1$/DNA hetero-complex structure, suggesting that the binding affinities for each N-terminal domain to the DNA are not equal, leading to an asymmetry of the binding pose. Notably, the putative DNA-binding residue Arg73 can clearly be resolved in monomer A but not in monomer B. Likewise, the density of the wing motif in monomer A is nestled within the minor groove, where Asn89 would be well placed to interact with DNA—an observation that aligns with the reported function of the wing motif to provide increased specificity[25]. In contrast, this motif in monomer B is less well resolved and appears to exhibit a weaker interaction with the minor DNA groove. Collectively, we hypothesize that, in the presence of DNA, one monomer of NanR binds the operator to partially stabilize the hetero-complex, while the opposing monomer undergoes a conformational change to untwist the α4-helices before engaging the DNA. We believe the asymmetry in the DNA-binding pose is a prerequisite to accommodate further dimers of NanR, given the proximity we observe between each dimer within the NanR-dimer$_3$/DNA hetero-complex, as they span the entire (GGTATA)$_3$-repeat operator.

Collectively, our findings offer formal support for a mechanism of sialoregulon repression in *E. coli* (Fig. 8) that is unique among reported GntR-type regulator mechanisms. The combination of cooperative binding to a repeat DNA sequence, a process mediated by atypical N-terminal extensions of the DNA-binding domain, and the formation of a multimeric protein–DNA hetero-complex distinguish NanR from other reported modes of transcriptional regulation among the GntR superfamily. Importantly, we also functionally validate Neu5Ac as the allosteric modulator of NanR, which had previously been proposed but lacked formal supporting evidence at the molecular level. By defining the mechanisms of induction and of gene repression for NanR, our studies extend our knowledge of the GntR superfamily and our understanding of the complex interactions between protein and DNA that lie at the heart of many biological processes.

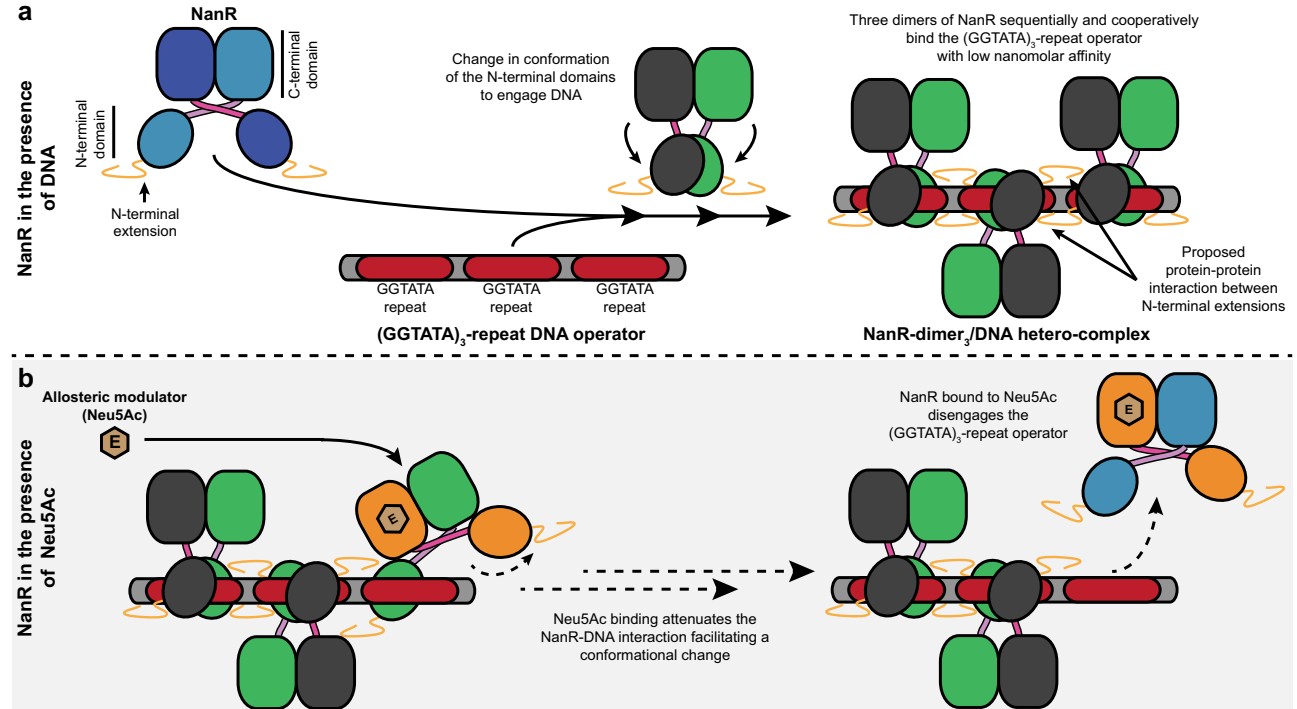

**Fig. 8 E. coli NanR regulation of gene expression in the sialoregulon.** Based on the data from this study, a schematic of the proposed mechanism for the regulation of gene expression by NanR is provided. **a** To repress gene expression, dimers of NanR cooperatively and with nanomolar affinity bind to each of the three GGTATA repeats to form a NanR-dimer₃/DNA hetero-complex through rearrangement of their N-terminal DNA-binding domains. This cooperative assembly is believed to be mediated by an N-terminal extension, unique to NanR among closely related GntR-type regulators. **b** The binding of the allosteric modulator Neu5Ac (beige hexagon) to the C-terminal effector-binding domain of NanR triggers a conformational change that attenuates the protein–DNA interaction. This facilitates a conformational change that results in NanR disengaging from the (GGTATA)₃-repeat operator, relieving repression of gene expression.

## Methods

**Protein cloning and expression.** *E. coli* K12 *nanR* (UniProt accession—P0A8W0; Supplementary Table 10) was commercially synthesized by GenScript and sub-cloned into expression vector pET28a. A truncated *nanR* fragment (NanR³³⁻²⁶³) was amplified using specific primers (Supplementary Table 10) and cloned into pET28a using an In-Fusion HD Cloning Kit (Takara Bio USA) as per the manu-facturer's instructions. The resulting recombinant plasmid was transformed into chemically competent *E. coli* BL21(DE3) cells, which were then cultured in Luria-Bertani growth medium supplemented with kanamycin (30 μg mL⁻¹) and ZnCl₂ (100 μM) at 37 °C with shaking at 220 rpm to an OD₆₀₀ of ~0.6. Protein expression was induced by the addition of isopropyl β-D-1-thiogalactopyranoside to 1 mM and the temperature was lowered to 26 °C for incubation overnight.

**Protein purification.** All purification steps for NanR and NanR³³⁻²⁶³ were con-ducted at 4 °C. Cells were harvested by centrifugation (Sorvall LYNX 4000 Superspeed) at 7000 × g for 10 min. Cell pellets were resuspended in lysis buffer (20 mM Tris-HCl (pH 8.0), 150 mM NaCl, 100 μM ZnCl₂), supplemented with cOmplete protease inhibitor cocktail (Roche), and lysed by sonication (Hielscher UP200S Ultrasonic Processor). Cell debris and insoluble material was pelleted by centrifugation at 32,000 × g for 30 min. As an initial purification step, protein was precipitated with 40% (w/v) ammonium sulfate for 1 h at 4 °C. Protein was pelleted at 11,000 × g for 15 min and resuspended in buffer A (20 mM Tris-HCl (pH 8.0), 50 mM NaCl), while the supernatant was discarded. The resuspended sample was dialyzed overnight in buffer A. NanR was then purified using a three-step proce-dure: anion exchange, heparin affinity, and size-exclusion chromatography using the ÄKTApure chromatography system (Cytiva). First, the dialyzed sample was applied to a HiTrap Q FF column (Cytiva) and washed with buffer A. Bound protein was eluted using a continuous gradient of buffer B (20 mM Tris-HCl (pH 8.0), 1 M NaCl). Fractions containing protein were identified by sodium dodecyl sulfate-polyacrylamide gel electrophoresis (SDS-PAGE) and subsequently pooled. The pooled sample was then applied to a HiTrap Heparin HP column (Cytiva), and the bound protein was eluted using a continuous gradient of buffer B. The eluted protein was pooled, concentrated via centrifugal ultrafiltration (30 kDa molecular weight cutoff; Sartorius), and loaded onto a Superdex 200 Increase 10/300 GL column (Cytiva) in buffer C (20 mM Tris-HCl (pH 8.0), 150 mM NaCl) and NanR was eluted as a single peak. The final purity was estimated to be approximately 95%, as highlighted by a single band on SDS-PAGE gels

(Supplementary Fig. 1i). Protein that was not immediately used was flash-frozen in liquid nitrogen and stored at −80 °C.

**Double-stranded DNA formation.** Complementary DNA oligonucleotides (Inte-grated DNA Technologies) were resuspended in a buffer consisting of 20 mM Tris-HCl (pH 8.0) and 150 mM NaCl, mixed at equimolar concentrations, and then hybridized by heating to 95 °C for 5 min, followed by cooling slowly to room temperature. DNA oligonucleotides used in EMSA and single-wavelength analy-tical ultracentrifugation experiments were FAM-5′ labeled on both strands to improve sensitivity. Double-stranded DNA oligonucleotides were stored at −20 °C until use.

**Electrophoretic mobility shift assays.** Double-stranded FAM-5′-labeled DNA oligonucleotides were diluted to 10 nM in gel shift buffer (10 mM MOPS (pH 7.5), 50 mM KCl, 5 mM MgCl₂, and 10% (v/v) glycerol). Twelve-well Novex 6% Tris-glycine gels (Invitrogen) were pre-run in 0.5× Tris-Borate-EDTA (TBE) buffer (40 mM Tris-HCl (pH 8.3), 45 mM boric acid, and 1 mM EDTA) at 200 V for 30 min at 4 °C. Protein and DNA oligonucleotides were mixed and incubated at room temperature for 30 min to allow samples to reach equilibrium. Electro-phoresis was performed immediately on the pre-run gels in 0.5× TBE buffer at 200 V for 20 min at 4 °C. Following electrophoresis, gels were imaged using a Typhoon FLA 9500 Biomolecular Imager (Cytiva) with a 473-nm excitation source and a long-pass emission filter or a ChemiDoc MP (BioRad).

**Analytical ultracentrifugation using fluorescence optics.** To assess protein–DNA interaction, fluorescence-detected sedimentation velocity experi-ments were performed in a Beckman Coulter Model XL-A analytical ultra-centrifuge using double sector epon-charcoal centerpieces fitted with sapphire windows in an An-50 Ti eight-hole rotor at 20 °C. NanR was titrated against a 5′-FAM-labeled (GGTATA)₃-repeat oligonucleotide (80 nM) at 11 protein con-centrations (2-fold dilutions from 794 to 0.78 nM) in buffer C. Experiments in the presence of Neu5Ac were obtained using buffer C, supplemented with 20 mM Neu5Ac and an analogous set-up as above. We frequently observe a minor com-ponent at 1 S (approximately 2% of the signal) in the DNA samples, which is likely a small amount of single-stranded DNA. We verified that NanR did not interact with the 5′-FAM label (Supplementary Fig. 1d); when mixing free FAM with NanR

and monitoring sedimentation at 495 nM, FAM does not sediment with NanR. Data were collected at 50,000 rpm, where sedimentation was monitored using the fluorescence emission optical system (AVIV Biomedical). To generate an artificial bottom, 50 μL of FC43 fluorinert oil was loaded into the bottom of each cell. A radial calibration was performed prior to each experiment at 3000 rpm using a calibration cell containing 10 μM fluorescein in 10 mM Tris-HCl (pH 7.8) and 100 mM KCl. Photo-multiplier tube (PMT) voltage and gain were adjusted for each cell, while an appropriate focusing depth was selected to maximize the signal and minimize the inner filter effect for the highest NanR concentration. A PMT voltage and gain setting of 58% was used across all cells. To assess the oligomeric structure of NanR and the effect of Neu5Ac in solution, sedimentation velocity experiments were performed using a Beckman Coulter Model XL-I analytical ultracentrifuge with the same set-up as described above. Data were obtained at 50,000 rpm using the absorbance optical system at 280 nm, measuring protein at three different concentrations (3.3, 10, and 30 μM) in buffer C. Experiments were repeated in buffer C, supplemented with 20 mM Neu5Ac, and in buffer C, supplemented with fluorescein (3 μM) (Thermo Fisher Scientific), which served as a FAM and protein only control. All above data were analyzed using SEDFIT[45]. Sedimentation data were fitted to either a continuous size distribution [c(s)] or a continuous mass distribution [c(M)] model. Fit data are presented using GUSSI[46]. The buffer density, buffer viscosity, and an estimate of the partial specific volume of the protein sample based on the amino acid sequence were also determined using SEDNTERP.

**Analytical ultracentrifugation using absorbance optics**. To test the effect of the N-terminal truncation on DNA binding, sedimentation velocity data were obtained at 50,000 rpm, with the same set-up as described above, using the absorbance optical system at 495 nm to monitor the sedimentation of the FAM-5′-labeled (GGTATA)₃-repeat oligonucleotide (3 μM) when titrated against NanR was titrated (3, 12, and 24 μM). Experiments were repeated with NanR[33–263] and DNA using the same concentrations as the wild-type protein. All data were analyzed using UltraScan 4.0, release 2578[47]. Sedimentation data were evaluated according to methods reported earlier[48]. Briefly, 2DSA[31] is used to remove systematic time and radially invariant noise contributions to the data and to fit the boundary conditions of the sample column. Monte Carlo analysis[49] is used to estimate the effect of stochastic noise on the obtained hydrodynamic parameters (sedimentation coefficient, diffusion coefficient). The buffer density, buffer viscosity, and an estimate of the partial specific volume of the protein sample based on the amino acid sequence were determined using UltraScan[47].

**Determining the dissociation constant of the NanR–DNA interaction**. The apparent affinity of the NanR–DNA interaction was measured by comparing the ratio of NanR-bound and unbound FAM-5′-labeled DNA oligonucleotide in both the EMSA and analytical ultracentrifugation experiments. This ratio was determined from the fluorescence-detected sedimentation velocity data by the integration of the peaks in the c(s) distribution, where a shifted species relative to the DNA signal represented hetero-complex formation. In the EMSA, the ratio of unbound versus bound DNA was determined by densitometry using ImageJ[50]. All further data analysis was performed using Prism 8 (GraphPad Software Inc.). When the fraction containing bound DNA was plotted against NanR concentration, the data were best explained by the Hill model (Eq. 1) with an Akaike information criterion (AIC) value of 99%, when compared to a non-cooperative binding model (AIC value of 1%).

$$\theta = \frac{[L]^n}{K_D + [L]^n} \tag{1}$$

Here, $\theta$ is the fraction of the DNA that is bound by NanR, $[L]$ is the concentration of bound NanR, $K_D$ is the apparent dissociation constant, and $n$ is the Hill coefficient.

**Bioinformatics**. To identify and compare NanR protein sequences, a sequence homology search within the Protein Data Bank (PDB) was performed using the online basic local alignment search tool (BLAST) program BLASTp[51]. Amino acid sequences of known GntR protein homologs were then sourced from the UniProt database and the PDB, respectively. Using these sequence homologs, a multiple sequence alignment was performed using Clustal Omega[52] and an image was generated using ESPript 3.0[53]. The disorder probability for NanR was estimated using the RONN (https://www.strubi.ox.ac.uk/RONN) and PrDOS (http://prdos.hgc.jp/cgi-bin/top.cgi) online servers (Supplementary Fig. 5c).

**Analytical ultracentrifugation with multiwavelength detection**. Multiwavelength sedimentation velocity is an emerging strategy to characterize complex mixtures by deconvoluting the spectral signals of the interaction partners into separate sedimentation profiles. Because it is a relatively new technique, we include an overview here (Supplementary Fig. 3). Briefly, the intrinsic extinction profile of each interacting partner is used to deconvolute the hydrodynamic data, collected over a range of wavelengths (e.g., 220–300 nm), into separate sedimentation profiles for each component. The data are then scaled to molar concentrations[54,55]. This is easily achieved when the intrinsic extinction profile for the interacting

partners is sufficiently different, for example, when comparing protein and DNA spectra[54]. Once deconvoluted and on a molar scale, the stoichiometry of the complex can simply be extracted by integrating the molar ratio of the co-migrating peaks[56]. Thus, multiwavelength sedimentation velocity experiments provide both hydrodynamic and spectral characterization of an interacting system to define the stoichiometry of association, as well as the hydrodynamic properties such as the mass and frictional ratio of each species.

Multiwavelength sedimentation velocity experiments were performed in a Beckman Coulter Optima analytical ultracentrifuge using double sector epon-charcoal centerpieces fitted with sapphire windows in an An-60 Ti four-hole rotor at 20 °C. Samples were prepared with increasing loading concentrations of NanR (0.5, 1.5, 3, and 5 μM) with respect to (GGTATA)₃-repeat DNA (0.5 μM) in 50 mM sodium phosphate (pH 7.4) and 150 mM NaCl. Data were collected at either 50,000 or 60,000 rpm and sedimentation was monitored using the ultraviolet absorption system in intensity mode, scanning only a single cell. Sedimentation velocity scans were recorded in the range of 220–300 nm with 2 nm increments, providing 41 individual datasets for each loading concentration. All data were analyzed using UltraScan 4.0[47]. Initially, multiwavelength sedimentation velocity datasets from each wavelength were analyzed using 2DSA[31] to remove systematic noise components and to determine boundary conditions of the sample column as reported above. Iteratively refined 2DSA models from each wavelength were used to generate a sedimentation profile for each wavelength mapped to a common time grid spectral deconvolution of the multiwavelength data using the molar extinction coefficient profiles of each spectral contributor generates the hydrodynamic results for each contributor. The partial specific volume for NanR was predicted based on the amino acid sequence of NanR using UltraScan and by assuming a partial specific volume of 0.55 mL g⁻¹ for DNA and using the determined stoichiometry to calculate a weight average partial specific volume (see below). Buffer density and viscosity were determined based on the buffer composition (50 mM sodium phosphate (pH 7.4), 150 mM NaCl) using UltraScan. Phosphate was chosen as the buffer over Tris-HCl to minimize background absorbance and therefore maximize signal from the protein and DNA.

Molar extinction profiles were determined by performing a dilution series for both NanR and DNA by collecting an absorbance spectrum across the spectral range of interest (220–300 nm) using a Genesys 10s benchtop spectrophotometer (Thermo Fisher Scientific). The dilution series of each absorbance spectra was fitted to intrinsic extinction profiles as we and others have described previously[54,56,57]. The resulting intrinsic extinction profiles were scaled to molar concentration using an extinction coefficient of 13,980 M⁻¹ cm⁻¹ at 280 nm for wild-type NanR and for NanR[32–263] as calculated by ExPASy ProtParam from the amino acid sequence. For the (GGTATA)₃-repeat oligonucleotide, an extinction coefficient of 567,112 M⁻¹ cm⁻¹ at 260 nm was determined by the nearest-neighbor method[58]. The vector angle between these spectral profiles was found to be 63.3°, which represents good orthogonality between spectra and therefore ensure separability. An angle of 0° reflects linear dependence or perfect overlap, while an angle of 90° indicates perfect orthogonality or no spectral overlap. Next, the spectral profiles scaled to molar concentration were subsequently used to deconvolute the noise-corrected multiwavelength data into separate datasets for the NanR and DNA components using the non-negatively constrained least squares algorithm[55] as previously described[54,56,57]. These deconvoluted datasets were individually analyzed by the 2DSA method using UltraScan[31]. The resulting amplitudes of the deconvoluted species involved in hetero-complex formation were then integrated to directly provide the molar stoichiometry of the NanR–DNA hetero-complexes. A summary of these integration results is shown in Supplementary Tables 4 and 5.

To determine the molar mass of each species in solution, a weight-averaged partial specific volume was estimated for each complex using Eq. 2.

$$\bar{v} = \frac{M_1 \bar{v}_1 + M_2 \bar{v}_2}{M_1 + M_2} \tag{2}$$

Here, the molar mass measured in Daltons is required for the NanR ($M_1$) and DNA ($M_2$) components, along with the partial specific volume of the NanR ($\bar{v}_1$) and the DNA ($\bar{v}_2$). Molar masses of 59, 118, and 177 kDa was used for the NanR-dimer₁, NanR-dimer₂, and NanR-dimer₃ protein components, respectively. A molar mass of 21.5 kDa was used for the (GGTATA)₃-repeat oligonucleotide. The partial specific volume used for NanR ($\bar{v}_1$) was 0.7295 mL g⁻¹, while the partial specific volume used for the DNA ($\bar{v}_2$) was 0.55 mL g⁻¹.

**Differential scanning fluorimetry**. Differential scanning fluorimetric experiments were performed using the Prometheus NT.48 instrument (NanoTemper Technologies). NanR was prepared at 30 μM in buffer C, loaded into glass capillaries and placed into the sample holder. Detection was achieved through excitation of tryptophan residues within the protein at 280 nm, while the intrinsic fluorescence intensity was recorded at 330 and 350 nm. The laser intensity was adjusted to 16%, based on the number of tryptophan residues. Samples were heated from 20 °C to a 95 °C at a ramp rate of 1 °C min⁻¹, taking fluorescence readings at each time point. Duplicate measurements were performed for each sample, while experiments were repeated at the same protein concentration in buffer C supplemented with Neu5Ac (20 mM). Data analysis was performed using PR.ThermControl software (NanoTemper Technologies) where an apparent melting point ($T_m$) of each sample in °C was obtained by taking the first derivative of the 350/330 nm ratio.

**Isothermal titration calorimetry**. Calorimetric titrations of NanR with Neu5Ac were performed with a Nano Isothermal Titration Calorimeter (TA Instruments). Purified NanR was initially concentrated to a final concentration of 416 µM via centrifugal ultrafiltration (30 kDa molecular weight cutoff; Sartorius) and then extensively dialyzed against buffer C. Neu5Ac was prepared in the same buffer by diluting a 100 mM stock solution to a final concentration of 1 mM. Protein sample (200 µL) was loaded in the sample cell, and 50 µL of Neu5Ac was loaded into the injection syringe. Titrations were initiated by a 1 µL injection, followed by 24 consecutive 2 µL injections every 200 s at 8 °C and a constant stirring speed of 60 rpm. A blank correction was obtained by injection of Neu5Ac (1 mM) into buffer C using an identical set-up. Titration data were integrated using NITPIC[59,60] and analyzed in SEDPHAT by discarding the initial injection and fitting the binding isotherm 1:1 interaction model[61] to obtain $K_D$ values.

**Crystallization, phase determination, and structure refinement**. Despite extensive screening, initial crystals were of poor quality and, following data collection at the Australian Synchrotron MX2 beamline, these crystals diffracted to ~5 Å resolution. To overcome this, we performed in situ proteolysis with the addition of 10 µg mL$^{-1}$ chymotrypsin having predicted that the N-terminal extension is predominantly disordered (Supplementary Fig. 5c) and reasoning that this would aid crystallization. Prior to crystallization, the protein solution/protease mixture was incubated on ice for 30 min. The initial dataset collected at 13 keV had a positive anomalous correlation, indicating the presence of a metal. An elemental analysis of the NanR crystals was carried out using X-ray fluorescence, showing an emission peak at 8639 eV (Supplementary Fig. 5a), and a multiwavelength anomalous diffraction scan was performed around the Zn-absorption edge with a peak evident at 9670.10 eV (Supplementary Fig. 5b), consistent with the presence of zinc in the crystals. The presence of an intrinsically bound zinc ion was further supported using inductively coupled plasma mass spectrometry.

An initial 2.29 Å C-terminal domain substructure was solved using the single anomalous diffraction (SAD) method. Crystals were obtained at 8 °C using the sitting-drop vapor-diffusion method and in situ proteolysis by mixing 400 nL of NanR (20 mg mL$^{-1}$) in buffer C supplemented with 20 mM Neu5Ac, since its presence increased the thermal stability of the protein (Supplementary Fig. 4a), with 400 nL of reservoir solution containing 0.1 M Tris-HCl (pH 8.5), 0.2 M magnesium chloride hexahydrate, and 30% (w/v) PEG 4000. For data collection, the crystals were cryoprotected in the same reservoir solution supplemented with 15% (w/v) glycerol/ethylene glycol and then flash-frozen. At a wavelength of 1.2782 Å (remote from the edge), 22 datasets were collected at a detector distance of 245–255 mm, across 5 different crystal positions from a single crystal where diffraction ranged from 2.62 to 2.29 Å. For each dataset, 3600 frames were collected with an exposure of 0.1 s per frame, with an X-ray beam attenuation of 50%. These datasets were processed in XDS[62] displaying $I2_12_12_1$ symmetry and were then analyzed with XDS_NONISOMORPHISM[63] to identify the most isomorphous datasets. Based on this analysis, eight isomorphous datasets were selected, merged, and scaled using XSCALE[62], to improve the zinc anomalous signal. The crystal structure of the C-terminal domain was solved using the SAD protocol in the Auto-Rickshaw pipeline[64]. Input diffraction data were prepared and converted for use in Auto-Rickshaw using programs within the CCP4 suite[65]. FA values were calculated using the program SHELXC[65]. Based on an initial analysis of the data, the maximum resolution for substructure determination and initial phase calculation was set to 2.70 Å. Both heavy atoms requested were found using the program SHELXD[66]. The correct hand for the substructure was determined using the ABS program[67], while initial phases were calculated following density modification using SHELXE[66]. The initial phases were further improved using density modification and phase extension to 2.29 Å resolution using RESOLVE[68]. Fifty percent of the model was built using the program ARP/wARP[69]. The resulting model was improved by iterative manual building in COOT[70] and refinement using PHENIX[71] and included residues 120–247 of the C-terminal domain.

We subsequently solved a 2.10-Å dataset using a combination of molecular replacement with the C-terminal domain substructure and the SAD method. Crystals were obtained at 8 °C using the sitting-drop vapor-diffusion method and in situ proteolysis by mixing 400 nL of NanR (20 mg mL$^{-1}$) in buffer C, supplemented with 20 mM Neu5Ac and 400 nL of reservoir solution (0.1 M sodium HEPES (pH 7.5), 0.2 M sodium acetate trihydrate, and 25% (w/v) PEG 3350). Once crystals were flash-frozen, a single dataset was collected at a wavelength of 0.9537 Å over 180° and was processed in XDS[62] displaying $P2_1$ symmetry. Using the C-terminal domain substructure as a search model, Auto-Rickshaw was used for phase enhancement and model completion[64]. The resulting model was improved by iterative manual building in COOT[70] and refinement using PHENIX[71]. No density was visible for residues 1–30 and 247–263 in chain A or for residues 1–30 and 245–263 in chain B—presumably these were cleaved by proteolysis or disordered. We found differential electron density that interacts with the α4-helix connecting the N- and C-terminal domains. Two polyethylene glycol tails fit well into this electron density and PEG 3350 was present in the crystallization buffer (Supplementary Fig. 5e). The final model was validated using MOLPROBITY[72]. The dimer interface was analyzed using PDBePISA[73]. All structural graphics were prepared using PyMOL and UCSF Chimera[74]. All data collection and refinement statistics are summarized in Supplementary Table 7.

**Small-angle X-ray scattering (SAXS) analysis**. SAXS data were collected at the Australian Synchrotron SAXS/WAXS beamline using an inline co-flow size-exclusion chromatography set-up to minimize sample dilution and maximize signal-to-noise ratio[75]. Purified NanR at 10 mg mL$^{-1}$ (340 µM) was injected (70 µL) onto an inline Superdex S200 5/150 Increase (Cytiva), equilibrated with buffer C, and supplemented with the radical scavenger 0.1% (w/v) sodium azide, using a flow rate of 0.45 mL min$^{-1}$. To investigate the effect of Neu5Ac, the inline S200 column was re-equilibrated in buffer supplemented with 20 mM Neu5Ac. NanR-DNA hetero-complex was prepared by incubating NanR (340 µM) and (GGTATA)$_2$-repeat DNA (170 µM) on ice for 30 min prior to injection. 2D intensity plots were radially averaged, normalized against sample transmission, and background-subtracted using the Scatterbrain software package (Australian Synchrotron). The ATSAS software package (version 3.0) was used to perform the Guinier analysis (PrimusQT[76]) to calculate the pairwise distribution function $P(r)$ and the maximum interparticle dimension ($D_{max}$) and to evaluate the solution scattering against the structural models solved in this study (CRYSOL[77]). The molecular mass of each sample was estimated using the SAXS-MoW2 package[78] and from the Porod volume. All data collection and processing statistics are summarized in Supplementary Table 8.

**Single-particle cryo-EM sample preparation**. NanR-dimer$_1$/DNA hetero-complex, was prepared by mixing NanR (17 µM) and (GGTATA)$_2$-repeat DNA (8.5 µM). NanR-dimer$_3$/DNA hetero-complex was prepared by mixing NanR (85 µM) and the (GGTATA)$_3$-repeat operator DNA (8.5 µM). To reduce sample heterogeneity and remove aggregates, each hetero-complex was purified using size-exclusion chromatography (Supplementary Figs. 8a and 10a, respectively). Following equilibration for 1 h at 4 °C, the sample was loaded onto a Superdex 200 Increase 10/300 GL column (Cytiva) pre-equilibrated with buffer A. Fractions consistent with hetero-complex formation were pooled and diluted to a final concentration of 0.5 mg mL$^{-1}$. Frozen-hydrated samples were prepared on plasma-cleaned Quantifoil R1.2/1.3 holey carbon EM grids (Quantifoil) using a Vitrobot Mark IV vitrification robot (FEI) with a 3-s blotting time, 100% humidity, and −3 mm blotting offset.

**Single-particle cryo-EM data acquisition**. For the NanR-dimer$_1$/DNA hetero-complex, automated data acquisition was performed using a Titan Krios™ electron microscope (FEI) at 300 kV, equipped with a K2 Summit™ direct detector (Gatan) and a GIF Quantum energy filter (Gatan). Cryo-EM imaging was performed using nanoprobe EFTEM zero loss imaging mode with a 20-eV slit width. A C2 Condenser aperture size of 50 µm and an objective aperture size of 70 µm were used during the imaging. At a nominal magnification of ×215,000, a magnified pixel size of 0.68 Å was provided. Movies were recorded using a K2 Summit™ direct detector (Gatan) operated in counting mode at a dose rate of $2e^-$ pixel$^{-1}$ s$^{-1}$. Each movie was a result of 12.8-s exposure with a total accumulated dose of 60 $e^-$ Å$^{-2}$, which were fractionated into 32 frames. The EPU software package (Thermo Fisher Scientific) was used for automated data collection and autofocus was set to achieve a defocus range from −0.5 to −2.5 µm.

For the NanR-dimer$_3$/DNA hetero-complex, automated data acquisition was performed using a Talos Artica™ electron microscope (FEI) at 200 kV, equipped with a Falcon III™ direct detector (FEI). A C2 Condenser aperture size of 50 µm and an objective aperture size of 100 µm were used during the imaging. At a nominal magnification of ×150,000, a magnified pixel size of 0.94 Å was provided. Movies were recorded using Falcon III™ direct detector (FEI) operated in counting mode at a dose rate of $0.8e^-$ pixel$^{-1}$ s$^{-1}$. Each movie had a total accumulated dose of 50 $e^-$ Å$^{-2}$, which was fractionated to 50 subframes. The EPU software package (Thermo Fisher Scientific) was used for automated data collection and autofocus was set to achieve a defocus range from −0.5 to −1.5 µm.

**Single-particle cryo-EM data processing and model building**. For the NanR-dimer$_1$/DNA hetero-complex, 3465 resulting movies were gain and motion corrected using MotionCor2[79] to output dose-weighted, beam-induced motion-corrected averages. CTF parameters were estimated on the corresponding non-dose-weighted averages using Gctf v1.06[80]. Both steps were performed using RELION v3.0[81]. A subset of images (544) was first used for automated particle picking using Gautomatch v0.53 with a defocus range −3 to −2 µm and a sphere diameter of 8 nm (Supplementary Fig. 7a). These particles were subsequently 2D classified in RELION v3.0[81]. The best 2D classes that showed clear structural details were used as a template for further automated particle picking. A total of 1,124,956 particles were then extracted from all 3465 dose-weighted movies, binned by 4 and subjected to 2 initial rounds of 2D classification ignoring CTF until first peak to filter out noisy/junk particles. The first round of 2D classification retained 695,465 particles of the hetero-complex, and a subsequent round cleaned up the dataset to retain 270,370 particles that had good signal-to-noise ratio (Supplementary Fig. 7b). These particles were re-centered on refined coordinates, extracted un-binned, and imported to cryoSPARC v2[82] for generation of an initial 3D ab initio reconstruction (Supplementary Fig. 7c). The resultant 3D reconstruction was then used as a reference model for 3D auto-refinement in RELION v3.0[81]. The first round of auto-refinement resulted in a 6.9 Å reconstruction, which displayed strong secondary structural elements corresponding to the dimer interface at the C-terminal domain, whereas the putative DNA-binding

region was comparatively noisy and showed signs of overfitting (Supplementary Fig. 7d). 3D classification was performed (tau_fudge = 4) by applying a 30-Å low-pass filter on the reference model from the previous auto-refinement, forming six discrete classes (Supplementary Fig. 7e). Four of these 3D classes showed distinct protein–DNA-bound features, where class 2 was the best resolved to 7 Å resolution (Supplementary Fig. 7e, dash circle). Using class 2 as a reference model, a new refinement was initiated using all particles from the previous step. This resulted in an improved 3D reconstruction with a resolution of 4.9 Å in which the dimer interface was sufficiently resolved, yet the protein–DNA interface still showed signs of over-fitting (Supplementary Fig. 7f). Bayesian polishing using the resultant 3D reconstruction with further auto-refinement increased the resolution to 4.4 Å. CTF refinement did not lead to any further increase in resolution. Using the same 3D classification settings and the 4.4 Å model as a reference, a third round of 3D classification was performed, which resulted in two high-resolution classes (classes 2 and 6), comprising 141,663 particles (Supplementary Fig. 7g). Using model 6 as the reference model internally low pass filtered to 12 Å, further auto-refinement resulted in an improvement to 4.2 Å resolution (Supplementary Fig. 7h). The DNA-binding region was further restricted by deriving a soft mask to eliminate the flexible terminal regions of the DNA oligonucleotide (Supplementary Fig. 7i). When applied in a masked refinement, this further improved the density of the protein–DNA interface. A final Bayesian polishing, restricting to first 20 frames, and auto-refinement teased out the signal of the 70.5 kDa NanR-dimer$_1$/DNA hetero-complex to a resolution of 3.9 Å (gold standard Fourier shell correlation (FSC) = 0.143 criteria) (Supplementary Fig. 7j). The final reconstruction encompasses density corresponding to the NanR dimer and 30 nucleotides (total mass of 59.7 kDa).

2D classification of the NanR-dimer$_1$/DNA hetero-complex dataset revealed class averages with clear density for DNA and distinct N- and C-terminal regions. However, 2D classification also revealed that, despite having areas of sufficiently thin ice, which resulted in high signal-to-noise ratio class averages, the sample also suffered from orientation bias. Further, only 20.37% of the initial 2D classified particles (695,465) contributed to structurally homogenous 3D classes (Supplementary Fig. 7g), which were ultimately used to determine the 3D reconstruction to 3.9 Å resolution. The FSC plot is shown in Supplementary Fig. 8b. The local resolution map estimated a range of resolution from 3.71 to ~5 Å (Supplementary Fig. 8c). The Euler angle distribution plot shows the extent of orientation bias (Supplementary Fig. 8e).

For the NanR-dimer$_3$/DNA hetero-complex, motion correction, CTF estimation, and template-based particle picking using Gautomatch v0.53 were performed as described for the NanR-dimer$_1$/DNA hetero-complex. All further processing was done using cryoSPARC v2[82]. Initial 2D classification yielded 211,384 particles from 2287 micrographs (Supplementary Fig. 9a). These were then further 2D classified to yield 141,501 particles showing clear density corresponding to the NanR-DNA hetero-complex (Supplementary Fig. 9b). Two distinct populations could be identified from the class averages readily: population 1 (Supplementary Fig. 9c) and population 2 (Supplementary Fig. 9d). Ab initio reconstruction was performed on population 1, which generated a 3D reconstruction of the NanR-dimer$_3$/DNA hetero-complex to a resolution of 8.3 Å (gold standard FSC = 0.143 criteria), as estimated by RELION v3.0[81] (Supplementary Fig. 9e). Further masked refinement strategies to tease out adjacent dimer–dimer interactions proved futile due to fewer number of particles in the pertinent class (Supplementary Fig. 9c). For population 2, the limited particle number as well as the comparatively weaker signal of the resultant class averages (Supplementary Fig. 9d) when compared to population 1 resulted in a 3D reconstruction that was not suitable for further processing and map interpretations (Supplementary Fig. 9f). The FSC plot is shown in Supplementary Fig. 10b. The Euler angle distribution plot shows the extent of orientation bias (Supplementary Fig. 10c).

The crystal structure of NanR was used as the reference model, while the (GGTATA)$_2$-repeat DNA oligonucleotide was modeled using the 3D-DART server[83]. The reference model was initially fit into the cryo-EM map for the NanR-dimer$_1$/DNA hetero-complex using Cryo_fit within PHENIX[71] and then further refined using MDFF in ISOLDE[84] to generate a secondary structural model. The resulting model was then improved by iterative manual building in COOT[70] and refinement in PHENIX[71] using reference model restraints, followed by further rotamer and Ramachandran restraints. A map threshold in COOT[29] of 0.0056 was used to aid tracing of the flexible loops and residues in the N-terminal extension of NanR. Refinement was guided by MOLPROBITY[72] statistics. The NanR dimer coordinates from the previous cryo-EM model, along with the (GGTATA)$_3$-repeat DNA oligonucleotide (modeled from 3D-DART[83]), were rigid body docked using UCSF Chimera[74] to generate the NanR-dimer$_3$/DNA hetero-complex model. All structural graphics were prepared using UCSF Chimera[74]. All data collection and refinement statistics are summarized in Supplementary Table 9.

**Reporting summary**. Further information on research design is available in the Nature Research Reporting Summary linked to this article.

## Data availability

The data that support this study are available from the corresponding author upon reasonable request. The atomic models for NanR bound to Neu5Ac, the NanR-dimer$_1$/DNA hetero-complex, and the NanR-dimer$_3$/DNA hetero-complex are available through the Protein Data Bank with the accession codes 6ON4, 6WFQ, and 6WG7, respectively.

Cryo-EM reconstructions of the NanR-dimer$_1$/DNA hetero-complex and NanR-dimer$_3$/DNA hetero-complex are available through the Electron Microscopy Data Bank with accession codes EMDB-21652 and EMDB-21661, respectively. Small-angle X-ray scattering data for NanR, NanR in the presence of Neu5Ac, and the NanR-dimer$_1$/DNA hetero-complex are available through the Small Angle Scattering Biological Data Bank with accession codes SASDHR9, SASDHS9, and SASDHT9, respectively. The *nanR* gene is available through the UniProt database with the accession code (PA08W0). Source data are provided with this paper.

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

## Acknowledgements

We thank staff at the Australian Synchrotron MX2 and SAXS/WAXS beamlines for their assistance in data collection and the New Zealand Synchrotron Group for enabling access; Janet Newman (Collaborative Crystallisation Centre) for assistance with protein crystallization, Yee-Foong Mok (Bio21 Institute, University of Melbourne) for his assistance with analytical ultracentrifugation experiments; and Tim Cooper (Massey University) for critical reading of the manuscript. We are grateful to the Biomolecular Interaction Centre (University of Canterbury), the Canterbury Medical Research Foundation, and the Maurice Wilkins Centre for scholarship support to C.R.H. We acknowledge the New Zealand Royal Society Marsden Fund (to R.C.J.D., UOC1506), Ministry of Business, Innovation and Employment Smart Ideas grant (to R.C.J.D., UOCX1706), ARC LIEF grants (to G.R., LE120100090 and LE200100045), Swedish Governmental Agency for Innovation Systems (to R.F., 2017-00180), Centre for Antibiotic Resistance Research (CARe) at University of Gothenburg (to R.F.), NHMRC grants (to J.M.M., 1172929 and 9000653), and the Victorian Government Operational Infrastructure Support Scheme (to J.M.M.). Funding to B.D. from NIH for grant and UltraScan multiwavelength development support (GM120600 and NSF-ACI-1339649), NSF/XSEDE grant for support towards UltraScan supercomputer calculations (TG-MCB070039N), and University of Texas grant (TG457201) is acknowledged. This research was undertaken in part using the SAXS/WAXS and MX2 beamlines at the Australian Synchrotron, part of ANSTO, and made use of the ACRF detector; the Monash Ramaciotti Centre for Cryo-Electron Microscopy (a node of Microscopy Australia) and made use of the Multi-modal Australian ScienceS Imaging and Visualisation Environment (www.massive.org.au) for data processing; and the Canadian Center for Hydrodynamics, University of Lethbridge for MWL-SV experiments, with support from the Canada Foundation for Innovation Grant CFI-37589 (to B.D.).

## Author contributions

C.R.H., R.A.N., and R.C.J.D. conceived the project. C.R.H. designed and performed experiments and analyzed data with support in data interpretation from E.B., J.M.M., and R.F. H.V. performed cryo-EM imaging experiments with assistance from G.R. S.P. determined the crystal structure. A.H., M.D.W.G., and B.D. contributed to analytical ultracentrifugation experiments. D.M.W. performed isothermal titration experiments. C.R.H., H.V., and R.C.J.D. co-wrote the paper. R.C.J.D. supervised the project. All authors commented on the manuscript.

## Competing interests

The authors declare no competing interests.
