## [Peer Review File · Nature Communications]

Reviewers' Comments:

Reviewer #1:

Remarks to the Author:

Horne, Venugopal *et al.* have elucidated the molecular mechanism by which the bacterial transcription factor NanR represses genes implicated in the sialic acid metabolism in *E. coli*. The authors have used a wealth of biochemical and biophysical methods to perform a comprehensive study of the interaction of NanR with conserved DNA operator sites and its regulation by the effector Neu5Ac. Particularly, they solved the first X-ray structure of a Neu5Ac-bound NanR complex. Impressively, they were able to use cryo-EM to solve near-atomic to medium resolution structures of NanR bound to DNA, an extremely flexible and relatively small molecular complex. Altogether, the different structures provide valuable insights into the molecular choreography of NanR binding to DNA and help rationalize the biochemical data. Overall, the work by Horne, Venugopal and colleagues is well executed, well written, and furnishes our understanding of the sialic acid metabolism in bacteria. This manuscript would definitely attract a broad readership and should be published in Nature Communications provided that the following points are addressed.

Major points:

1. The manuscript provides multiple evidences that Neu5Ac binds to the ligand-binding domain (LBD) of NanR which in turn affects the ability of NanR to bind DNA. However, the stoichiometry by which Neu5Ac binds to NanR2 (Neu5Ac1-NanR2 or Neu5Ac2-NanR2) remains elusive and is not discussed in the manuscript. The Neu5Ac-bound NanR2 crystal structure presented in this manuscript shows only one molecule of Neu5Ac in the LBD of one monomer while the other monomer remains in apo form. According to the authors, the presence of Neu5Ac in the LBD triggers conformational changes that lock the DBD against the LBD of the other monomer, preventing it from binding DNA. With this reasoning, a Neu5Ac2-NanR2 complex (where both LBD are occupied) could not bind DNA at all. However, the authors showed that Neu5Ac reduces by only 2-fold the affinity of NanR for DNA but it does not abolish binding to DNA. Does this data imply that only one LBD site is occupied by Neu5Ac? Could the authors comment on that?

Along those lines, the authors confirm that the Neu5Ac induces a compaction of the overall structure in solution using SAXS. By fitting the scattering data with the asymmetric Neu5Ac-bound structure (Neu5Ac1-NanR2), the authors make the assumption that only one molecule of Neu5Ac is bound to the dimer in solution. But, in figure 8, the authors present a model in which two Neu5Ac molecules bind a dimer of NanR, removing it from the DNA. This contradicts their biochemical data which shows, in presence of the effector, a reduced affinity of NanR for DNA not a complete binding abolishment. These discrepancies need to be clarified in the discussion.

Addressing the question of the stoichiometry will greatly benefit to the understanding of the mechanism by which Neu5Ac attenuates the binding of NanR2 to DNA. The authors already possess data that they could use to obtain such information without the need to perform new experiments: 1) the stoichiometry can be derived from the ITC data, 2) a fit of a symmetric Neu5Ac2-NanR2 model on the Neu5Ac-bound NanR experimental SAXS data could be compared to the asymmetric model fit. Another alternative would be to perform native mass spectrometry on the Neu5Ac-bound NanR-dimer, should this equipment be at hand of the authors. If the stoichiometry still cannot be characterized, the authors should at least discuss it in the manuscript.

2. For the cryo-EM study of NanR-dimer1/DNA, the authors used a repeat of the operator consensus sequence (GGTATA)₂, although they solved the structure of a single NanR-dimer1 bound to only one operator sequence. What is the rationale for using two effectors instead of one? It adds heterogeneity, flexibility and complicates the data processing. This data heterogeneity is even visible in the 2D classes in Supplementary Fig. 7 where one can observe 2 NanR-dimer on the same DNA. I suspect

this is due to the low-affinity binding of NanR when working with only one operator sequence. The authors should explain their rationale to better guide the readers.

3. In the cryo-EM structure of NanR-dimer1/DNA, the resolution of the map is not sufficient to distinguish individual nucleotides nor to confidently assign the nucleotides sequence (5Å local resolution for the DBD and DNA). Moreover, the map has been masked during refinement to remove extra bits of flexible DNA. The authors should explain their reasoning for the confident assignment of the DNA sequence since they propose potential binding of amino acid residues to specific DNA base-pairs of the operator. The same interrogation applies for the NanR-dimer3/DNA. Also, highlighting the operator sequence in the model of Fig. 5 could improve the general readability, such as displayed in Fig. 6e.

4. Along similar lines, the authors point out particular amino acid residues (Arg59, Arg69, Arg73 and Asn89) possibly making sequence-specific contacts with the DNA bases within the operator sequence. These residues were identified based on a sequence alignment and a 3D alignment with the DNA-bound FadR structure, but the resolution of the present cryo-EM data does not seem sufficient to clearly visualize and confidently place side chains into the EM map. At this point, the possible role of these amino acid residues in binding the operator remains uncertain. Intriguingly, a study of Kalidova et al. (J. Bacteriol 2013 - reference 22 in the present manuscript) have shown in vivo that the mutation of only two (Arg69 and Arg73) of the aforementioned residues affects the ability of NanR to repress gene expression, suggesting a different mechanism of NanR binding to its operator than FadR. In the light of these results and in combination with the structural data of the authors, it would be highly interesting to further discuss the possible function of these residues in the context of an asymmetric NanR/DNA interaction.

Minor points:

5. Overall, the figures 4, 5, 6 and 7 illustrate with clarity important features of the structure models. However, when it comes to the description of conformational changes between different states, some panels (e.g. Fig. 5d, Fig. 7a) are a bit messy and hard to follow. I would suggest the authors to provide additional movies of the overall structures and, if possible, a morphing between the different states described in Fig. 7.

6. Figure 2e: the legend says 0.6 μM NanR33-263 although the figure shows 1.8 μM NanR33-263.

7. Figure 4g: the panel shows that the DBD of the apo conformation has few interactions with the Neu5Ac-bound LDB, implicating a small number of residues. In contrast, the interface between the DBD of the Neu5Ac-bound conformation and the apo LBD, which is interpreted by the authors as "forming an extensive new interface locking the N-terminal domain in a closed conformation", is not shown. It would be interesting to depict and compare side by side these two different interfaces.

8. Figure 5f-h: the transparent rendering of the EM map is very faint. It is hard to visualize even on the computer screen. The chosen panel views do not really help visualize potential interaction of amino acids with specific DNA-bases, as proposed by the authors. Also, it is not really clear from the text nor the figures if the binding mode to DNA of each NanR DBD is exactly the same (a perfect symmetry, like the FadR structure) or asymmetric. This part could be further improved together with the comment I made to the authors in point 4.

9. Supplementary Fig 2: using PDB ID to annotate the homologs of NanR is not really informative. The authors could use the species and name of the proteins they identified as homologs and provide the PDB ID in the legend or in the Methods section.

10. Supplementary table 8: 1) the camera should be Gatan "K2" and not "V2". 2) The Ramachandran

statistics do not correspond to the ones provided in the PDB deposition report.

11. The authors provided a data processing pipeline for the cryo-EM study of the NanR-dimer1/DNA. It would be convenient for the reader to also provide one for the NanR-dimer3/DNA complex in the supplementary information. The authors should also explain why they stopped processing the data after the first 3D refinement of the NanR-dimer3/DNA complex in opposition to the NanR-dimer1/DNA complex for which they performed extensive 3D classification.

12. There is a discrepancy between the legend of Supplementary Fig 4b referring to the concentration of NanR (120 μM) used in the ITC experiment compared to the Methods section stating a NanR initial concentration of 172 μM . Is 120 μM after dialysis? Please, address this discrepancy.

13. Page 14, line 4: The content of Supplementary Fig. 5c does not explain the sentence from the main text.

14. Page 17, line 3: in the sentence "In contrast, when bound to DNA, the α 4-helices do not cross and, while the resolution does not permit a detailed examination of how these helices interact, we observed that the α 4-helix of monomer A (with Neu5Ac and Zn^{2+}) is longer and kinked compared with the α 4-helix of monomer B", it is not clear whether the authors describe the atomic model from the X-ray structure or the model refined in the cryo-EM structure. They mention " α 4-helix of monomer A (with Neu5Ac and Zn^{2+})" but the cryo-EM complex of NanR bound to DNA does not contain any Neu5Ac. Could you please clarify this paragraph?

15. Page 18, line 13: To my understanding, the content of Supplementary Fig. 8j does not really explain the sentence from the main text. Maybe the authors wanted to refer to the panel 8h?

16. Page 20, line 1: the content of Supplementary Fig. 8e shows the angular distribution of the NanR-dimer1/DNA complex and is not related to the sentence from the main text. I would suggest the authors to double check the matching between the main text and their references to the supplementary information.

To conclude, this is an impressive piece of work.

Reviewer #2:

Remarks to the Author:

The NanR transcriptional repressor regulates genes associated with sialic acid metabolism in *E. coli*. This paper employs biophysical approaches to define the affinity, cooperativity, and stoichiometry of NanR binding to a (GGTATA)₃-repeat model DNA operator sequence and presents the structures of free NanR and NanR-DNA complexes. A mechanism is proposed for regulation of NanR-DNA interactions by the effector N-acetylneuraminate where the ligand allosterically disrupts DNA binding. This is an interesting system from the perspective of allosteric regulation and authors have used an impressive array of methods to probe it. However, there are some problems with analysis and interpretation of the biophysical data that detract from the potential impact of this study.

1. The most significant issue is that the model for allosteric regulation of NanR by acetylneuraminate in Fig. 8 is based on shaky biophysical data. The K_d for acetylneuraminate binding to NanR is reported as 183 μM . This seems far too weak to be biologically relevant. This K_d was obtained by ITC using a protein concentration of only 120 μM such that the C-factor (the product of the binding constant and the receptor concentration) is less than 1 when it should be at least 5 for reliable fitting. Thus, the reported confidence intervals of 91 to 275 μM seem too narrow. Equally troubling, binding of acetylneuraminate to NanR reduces its affinity for DNA by less than a factor of two. This change in affinity is tiny and seems incompatible with the structural change in NanR induced by

acetylneuraminate that appears to render the protein incapable of binding DNA.

2. There are several problems with the analysis of the gel shift and AUC data. It is impossible to assign any of the peaks in the $c(s)$ distributions in Fig. 2c to "complexes." This is a reversibly interacting system in rapid equilibrium so that features larger than the peak for the free DNA represent reaction boundaries, not discrete species. Also, it is very difficult to see many of the distributions in the plot due to the choice of colors and the thick linewidths.

3. It is clear that NanR binds to the (GGTATA)₃ repeat DNA with positive cooperativity but Figure 2b throws out much of the interesting information present in the gel-shift (Fig. 2a) and sedimentation velocity (Fig. 2c) experiments to fit to an apparent K_d and a phenomenological Hill coefficient. The problem with this kind of analysis is exemplified by the fit shown in Fig. S1g which gives a Hill coefficient of 2.8 for binding of two NanR to (GGTATA)₂. This is impossible as the Hill coefficient must be less than or equal to the binding stoichiometry. All of the bound species are lumped together to determine "fraction bound." Instead, the data should be analyzed by fitting the individual band amplitudes in the gel-shift and the signal-average sedimentation coefficients for the sedimentation velocity data to a real model that incorporates cooperativity. Supplementary Table 2 contains "weight average" (really signal-average) sedimentation coefficient data corresponding to Fig. 2C but they do not fit. Alternatively, the sedimentation velocity data could be directly fit to a model.

4. There are several mistakes in the legend for Fig. 2. First, the Hill model is a mass-action binding model that incorporates cooperativity. The "mass action binding model" should be labeled as a noncooperative model. In Fig. 2e the concentration of NanR33-363 is labeled as 1.8 μM but is indicated as 6.0 μM in the caption. The statement in the caption "All plots are presented as $g(s)$ distributions against the molar concentration of monomeric NanR33-263" does not make sense. This same statement is found in the legend to Fig. 3.

5. Why is the analysis of the NanR33-363 construct in Fig. 2d performed at 3 μM DNA whereas the wild-type analysis was performed using fluorescence detection at 10 nM DNA? They should be done under the same conditions to make a valid comparison.

6. The integrations of the peak amplitudes for protein and DNA performed in figures 2e, 2f, 3b-d are based on the erroneous assumptions that the peaks correspond to species and they are baseline separated. Neither is true. The main peak in the distributions in Fig. 3e may correspond to a saturated species but that is not proven by the presence of free protein. The peak appears slightly shifted to the right relative to the peak in Fig. 3d. It would be necessary to go to a protein concentration where the peak stops shifting with concentration to verify that it represents a stable species, not a reaction boundary.

7. The description of the AUC methods is excessively long and Fig. S3 is unnecessary.

We thank the reviewers for their positive and constructive comments, which we have incorporated into our revised manuscript entitled “**Mechanism of NanR gene repression and allosteric induction of bacterial sialic acid metabolism**” by Horne, Venugopal *et. al.* These suggestions have greatly improved the clarity of our presentation. We have addressed each reviewer comment (shown in *black italics*) in our point-by-point response below, where we respond in **blue text**, with changes **highlighted in grey** in this letter and the main text document. In addition, we have made minor editorial changes, which are also **highlighted in grey**.

Reviewer 1

Horne, Venugopal et al. have elucidated the molecular mechanism by which the bacterial transcription factor NanR represses genes implicated in the sialic acid metabolism in E. coli. The authors have used a wealth of biochemical and biophysical methods to perform a comprehensive study of the interaction of NanR with conserved DNA operator sites and its regulation by the effector Neu5Ac. Particularly, they solved the first X-ray structure of a Neu5Ac-bound NanR complex. Impressively, they were able to use cryo-EM to solve near-atomic to medium resolution structures of NanR bound to DNA, an extremely flexible and relatively small molecular complex. Altogether, the different structures provide valuable insights into the molecular choreography of NanR binding to DNA and help rationalize the biochemical data. Overall, the work by Horne, Venugopal and colleagues is well executed, well written, and furnishes our understanding of the sialic acid metabolism in bacteria. This manuscript would definitely attract a broad readership and should be published in Nature Communications provided that the following points are addressed.

Major points:

1.1. *The manuscript provides multiple evidences that Neu5Ac binds to the ligand-binding domain (LBD) of NanR which in turn affects the ability of NanR to bind DNA. However, the stoichiometry by which Neu5Ac binds to NanR2 (Neu5Ac1-NanR2 or Neu5Ac2-NanR2) remains elusive and is not discussed in the manuscript. The Neu5Ac-bound NanR2 crystal structure presented in this manuscript shows only one molecule of Neu5Ac in the LBD of one monomer while the other monomer remains in apo form. According to the authors, the presence of Neu5Ac in the LDB triggers conformational changes that lock the DBD against the LDB of the other monomer, preventing it from binding DNA. With this reasoning, a Neu5Ac2-NanR2 complex (where both LBD are occupied) could not bind DNA at all. However, the authors showed that Neu5Ac reduces by only 2-fold the affinity of NanR for DNA but it does not abolish binding to DNA. Does this data imply that only one LBD site is occupied by Neu5Ac? Could the authors comment on that?*

Response: We thank the reviewer for prompting further discussion on this important point and agree that our biophysical data does indeed imply that only one effector-binding domain (LBD) is occupied by Neu5Ac. This is evidenced by our SAXS data in the presence of Neu5Ac, where the theoretical scatter of the asymmetric crystal structure (Neu5Ac₁-NanR₂) best fit the solution scatter ($\chi^2=6.6$; **Supplementary Fig. 6a**). Thus, the structural reorganization of one monomer in the presence of Neu5Ac, which we observe in the crystal structure would only weaken rather than abolish DNA binding and would therefore be more consistent with the small change we

observed by EMSA. While this small change in affinity may appear ineffective to induce gene expression, it would allow other systems, such as DNA methylation to facilitate induction. We have now amended our discussion on the allosteric mechanism to suggest that the NanR/DNA interaction is weakened in response to Neu5Ac binding and have corrected our schematic models in **Fig. 6c** and **Fig. 8** to show only one LBD occupied by Neu5Ac. The text on page 19, lines 459-465 now reads: Overall, our biophysical and structural studies support a mechanism of sialoregulon repression in *E. coli* (**Fig. 8**) whereby three NanR dimers cooperatively bind the (GGTATA)₃-repeat operator upstream of gene/s encoding the enzyme machinery needed to import and metabolize sialic acids. Tight cooperative binding is mediated by the N-terminal extensions of the DNA-binding domain. Neu5Ac binding to NanR results in a structural rearrangement of the N-terminal domain in one monomer, attenuating affinity for DNA and allowing systems, such as DAM, to methylate and block NanR from re-binding, thereby inducing gene expression.

1.2. *Along those lines, the authors confirm that the Neu5Ac induces a compaction of the overall structure in solution using SAXS. By fitting the scattering data with the asymmetric Neu5Ac-bound structure (Neu5Ac1-NanR₂), the authors make the assumption that only one molecule of Neu5Ac is bound to the dimer in solution. But, in figure 8, the authors present a model in which two Neu5Ac molecules bind a dimer of NanR, removing it from the DNA. This contradicts their biochemical data which shows, in presence of the effector, a reduced affinity of NanR for DNA not a complete binding abolishment. These discrepancies need to be clarified in the discussion.*

Response: We thank the reviewer for picking up this error. We have now corrected our schematic model of the Neu5Ac-bound structure in both **Fig. 6c** (highlighting the conformational change between our crystal and cryo-EM structures) and **Fig. 8** (model mechanism) to show only one Neu5Ac molecule bound to NanR (Neu5Ac₁-NanR₂) and a reduced affinity through the structural reorganization of one monomer in response to Neu5Ac binding. We have also corrected the text where the model in **Fig. 6c** is discussed to match these changes. The text on page 15, lines 354-359 now reads: Neu5Ac binding promotes an opposite conformation (**Fig. 6a**, beige structure), whereby the N-terminal domain of one monomer moves closer to the C-terminal domain of the opposing monomer, allowing new interactions to be formed (**Fig. 4g**; **Fig. 6c**). This would lock the Neu5Ac-bound structure in a conformation that would be unfavorable for DNA binding, reducing the affinity for the NanR-DNA interaction.

1.3. *Addressing the question of the stoichiometry will greatly benefit to the understanding of the mechanism by which Neu5Ac attenuates the binding of NanR₂ to DNA. The authors already possess data that they could use to obtain such information without the need to perform new experiments: 1) the stoichiometry can be derived from the ITC data, 2) a fit of a symmetric Neu5Ac₂-NanR₂ model on the Neu5Ac-bound NanR experimental SAXS data could be compared to the asymmetric model fit. Another alternative would be to perform native mass spectrometry on the Neu5Ac-bound NanR-dimer, should this equipment be at hand of the authors. If the stoichiometry still cannot be characterized, the authors should at least discuss it in the manuscript.*

Response: Thanks again to reviewer 1 for these constructive suggestions. We agree that discussion of the stoichiometry by which Neu5Ac binds to the NanR dimer will benefit the reader in understanding the mechanism of gene regulation by NanR. To investigate the stoichiometry by which Neu5Ac binds the NanR dimer, we first

repeated our ITC experiments using a higher protein concentration (416 μM) and a lower Neu5Ac concentration (1 mM) to optimize the C-factor in line with the query from reviewer 2. When fitted to a 1:1 binding site model, the data gave a K_D value of 16 μM with a 95% confidence interval of 7 to 25 μM , provided a C-factor of ~ 32.5 and gave an N-value of 0.52 (data has now been included in **Supplementary Fig. 4b**). The N-value is consistent with one Neu5Ac molecule bound per NanR (Neu5Ac₁-NanR₂). To support this stoichiometry, we followed the suggestion by the reviewer to compare the theoretical scatter of the asymmetric crystal structure (Neu5Ac₁-NanR₂) to the theoretical scatter of a symmetrical Neu5Ac-bound model (Neu5Ac₂-NanR₂) using our SAXS data collected of NanR in the presence of Neu5Ac. Indeed, the theoretical scatter of the asymmetric crystal structure (Neu5Ac₁-NanR₂) gave a lower χ^2 value ($X^2=6.6$; **Supplementary Fig. 6a**) than the symmetrical Neu5Ac₂-NanR₂ model ($X^2=11.3$), which we have now included as a comparison in **Supplementary Fig. 6a**. Taken together, our crystallographic and biophysical studies suggest that only one Neu5Ac molecule is bound to the NanR dimer in solution, which leads to a conformational change that partially disrupts, rather than completely abolishing the NanR-DNA interaction.

We have further discussed this mechanism of gene regulation, including the stoichiometry in the manuscript and have amended our schematic models (**Fig. 6c**; **Fig. 8**) to support this (in line with the reviewer's query from point 1.2). The text on page 8, lines 194-197 now reads: We measured the dissociation constant (K_D) for Neu5Ac binding to NanR using isothermal titration calorimetry (**Supplementary Fig. 4b**), yielding a K_D of 16 μM (95% C.I. 7-25 μM) and an N-value of 0.52, which is consistent with one Neu5Ac bound per NanR dimer.

The text on page 12, lines 273-281 now reads: Small angle X-ray scattering experiments comparing NanR alone and NanR in the presence of Neu5Ac show a decreased R_g (32.5 to 31.4 \AA) (**Supplementary Fig. 6a**; **Supplementary Table 7**), supporting the observation that Neu5Ac compacts the protein in the crystal structure. Further, the Neu5Ac-free scattering data best fit the extended symmetrical apo model ($X^2=3.7$ using CRY SOL; **Supplementary Fig. 6a**), while the scattering data in the presence of Neu5Ac best fit the compact asymmetric crystal structure ($X^2=6.6$ using CRY SOL; **Supplementary Fig. 6a**), rather than a symmetrical Neu5Ac-bound model ($X^2=11.3$ using CRY SOL; **Supplementary Fig. 6a**). This suggests that only one Neu5Ac molecule has bound NanR, which is consistent with the stoichiometry (N-value of 0.52) obtained from our isothermal titration calorimetry experiments (**Supplementary Fig. 4b**).

The text on page 18, lines 424-427 now reads: Consistent with this model, our binding studies demonstrate that NanR binds Neu5Ac with micromolar affinity with a stoichiometry of one Neu5Ac molecule per NanR dimer, which is consistent with our crystallographic and solution studies.

2. For the cryo-EM study of NanR-dimer1/DNA, the authors used a repeat of the operator consensus sequence (GGTATA)₂, although they solved the structure of a single NanR-dimer1 bound to only one operator sequence. What is the rationale for using two effectors instead of one? It adds heterogeneity, flexibility and complicates the data processing. This data heterogeneity is even visible in the 2D classes in Supplementary Fig. 7 where one can observe 2 NanR-dimer on the same DNA. I suspect this is due to the low-affinity binding of NanR when working with only one operator sequence. The authors should explain their rationale to better guide the readers.

Response: We chose the (GGTATA)₂ repeat oligonucleotide because NanR binds poorly to the DNA sequence with only one GGTATA repeat (**Supplementary Fig. 1e**). We reasoned that the low affinity would make identifying hetero-complex particles from free protein/DNA difficult in the cryo-EM experiments. Secondly, from our preliminary analytical ultracentrifugation experiments we observed a monodisperse species for the NanR-dimer₁/(GGTATA)₂ hetero-complex at a protein to DNA ratio of 2:1. This hetero-complex was purified using size-exclusion chromatography prior to cryo-EM sample preparation to reduce any free interacting partners. Not surprisingly, there was still some evidence of heterogeneity in the 2D class averages, which we believe is attributed to crowding in regions of the cryo-EM grid with the thinnest ice, where the increased local protein:DNA ratio may have driven the assembly towards the higher order NanR-dimer₂/DNA hetero-complex. We could compensate for this heterogeneity and the terminal flexibility from the (GGTATA)₂-repeat DNA oligonucleotide by using a refinement mask. We have taken the opportunity to further clarify this important point in the manuscript. The text on page 12, line 289-291 now reads: We used the (GGTATA)₂ repeat oligonucleotide to solve this structure (**Fig. 5b**) as NanR bound poorly to the oligonucleotide with only one GGTATA repeat (**Supplementary Fig. 1e**).

3. *In the cryo-EM structure of NanR-dimer1/DNA, the resolution of the map is not sufficient to distinguish individual nucleotides nor to confidently assign the nucleotides sequence (5Å local resolution for the DBD and DNA). Moreover, the map has been masked during refinement to remove extra bits of flexible DNA. The authors should explain their reasoning for the confident assignment of the DNA sequence since they propose potential binding of amino acid residues to specific DNA base-pairs of the operator. The same interrogation applies for the NanR-dimer3/DNA. Also, highlighting the operator sequence in the model of Fig. 5 could improve the general readability, such as displayed in Fig. 6e.*

Response: We agree that the resolution is insufficient for assigning individual nucleotides or the sequence and as such have not attempted to do so from the cryo-EM data. Instead, we suggest putative DNA-binding residues based upon on sequence alignment with FadR, the mutational analysis performed by Kalivoda *et al.* 2013 (reference 22 in manuscript) and the chemistry (i.e. positive charge) of respective amino acid residue. That said, the reconstruction of the DNA oligonucleotide in the cryo-EM density for the NanR-dimer₁/(GGTATA)₂ dataset is unambiguously guided by the major and minor grooves of DNA. This same principle was applied to the NanR-dimer₃/(GGTATA)₃ dataset, along with the new knowledge of the DNA binding mode that we observed in the lower order hetero-complex. As such, we have now amended the text to clarify this statement and to reinforce our rationale on the placement of the (GGTATA)₂-repeat DNA oligonucleotide in the NanR-dimer₁/(GGTATA)₂ cryo-EM structure (**Fig. 5**) and the NanR-dimer₃/(GGTATA)₃ cryo-EM structure (**Fig. 6**). The text on page 13, line 313-314 now reads: Reconstruction of the DNA oligonucleotide in the cryo-EM density for this dataset is unambiguously guided by the major and minor grooves of DNA (**Fig. 5a**).

The text on page 16, line 377-379 now reads: Analogous to the NanR-dimer₁/DNA hetero-complex dataset, reconstruction of the (GGTATA)₃-repeat oligonucleotide in the cryo-EM density was unambiguously guided by the grooves of DNA.

Furthermore, we have included the (GGTATA)₂-repeat DNA oligonucleotide as an additional panel in Figure 5 (Fig. 5b), which is analogous to Fig. 6e to better guide the reader, and we are grateful to the reviewer for this helpful suggestion.

4.1. Among similar lines, the authors point out particular amino acid residues (Arg59, Arg69, Arg73 and Asn89) possibly making sequence-specific contacts with the DNA bases within the operator sequence. These residues were identified based on a sequence alignment and a 3D alignment with the DNA-bound FadR structure, but the resolution of the present cryo-EM data does not seem sufficient to clearly visualize and confidently place side chains into the EM map.

Response: The reviewer makes some excellent points. As discussed in comment 3 above, the putative DNA-binding residues we identify in this study (Glu58, Arg59, Arg69, Arg73 and Asn89) are based on a sequence alignment with FadR, the mutagenesis performed by Kalivoda *et al.* 2013 (reference 22 in manuscript), and side-chain chemistry (e.g. positive charge). The assignment of these residues is further supported by their proximity and orientation to the DNA, along with some side-chain density in the cryo-EM map (highlighted in Fig. 5g-i). For example, side-chain density is apparent in Arg73, which is located in the α -helix 3 (one of the best resolved regions of the overall cryo-EM map), while the other residues are less resolved, likely due to flexibility in their respective environments. To better visualize the density of these putative DNA-binding residues, we have reduced the transparency in Fig. 5g-i (this also addresses point 8 below from reviewer 1). In line with the reviewer's query, we have made clear in the manuscript that the resolution of the DNA-binding region (~ 5 Å) is indeed insufficient to resolve specific DNA base-pair contacts between these putative DNA-binding residues. Nevertheless, our assignment of these residues is reasonable given that they all satisfy the criteria summarized above. We have amended the text on page 13-14, lines 321-337 to reflect this: Based on our cryo-EM structure of the NanR-dimer₁/DNA hetero-complex, the above sequence comparison with FadR, a mutational analysis performed by Kalivoda *et al.* 2013²², and side chain chemistry (i.e. positive charge), we have identified nine putative DNA-binding amino acid residues. Ser33 in the α 1-helix, Glu58 in the α 2-helix, Gly68 and Ser71 in the α 3-helix, and Glu91 in the wing motif are likely to form an interaction with the phosphate backbone of DNA, while Arg59 in the α 2-helix, Arg69 and Arg73 in the α 3-helix, and Asn89 in the wing motif likely make sequence-specific contacts with the DNA bases within the operator sequence (Fig. 5f). There was a clear difference in local resolution between the two N-terminal domains. Comparatively, monomer A is better resolved, particularly in the wing motif and the α 3-helix (Fig. 5g-i, top panel), which allowed for several of the putative DNA-binding side chains, such as Arg73, to be assigned in the model. This suggests a difference in binding affinity between the N-terminal DNA-binding domains to the non-equivalent DNA binding sites. Despite the assignment of these putative DNA-binding residues based on these inferences, it is important to note that the resolution of the overall DNA-binding region (~ 5 Å) (Supplementary Fig. 8c) is insufficient to resolve specific DNA base-pair contacts with the (GGTATA)₂-repeat oligonucleotide. Nevertheless, this asymmetry in the DNA binding pose suggests there is a difference in binding affinity between the N-terminal DNA-binding domains and the non-equivalent DNA binding sites.

4.2. At this point, the possible role of these amino acid residues in binding the operator remains uncertain. Intriguingly, a study of Kalivoda *et al.* (J. Bacteriol 2013 - reference 22 in the present manuscript) have shown in

vivo that the mutation of only two (*Arg69* and *Arg73*) of the aforementioned residues affects the ability of *NanR* to repress gene expression, suggesting a different mechanism of *NanR* binding to its operator than *FadR*. In the light of these results and in combination with the structural data of the authors, it would be highly interesting to further discuss the possible function of these residues in the context of an asymmetric *NanR*/DNA interaction.

We thank reviewer 1 for prompting further discussion on this important point. We observe a higher local resolution in monomer A compared to monomer B, which meant we could more confidently assign putative DNA-binding residues to monomer A relative to monomer B (criteria for assignment is discussed above). This suggests to us that the DNA binding mode may be different for each monomer, which is not surprising since the DNA operator is non-palindromic. As an example, the putative DNA-binding residue, *Arg73* can clearly be resolved in monomer A, but not in monomer B. Furthermore, the density of the wing motif in monomer A appears to be nestled within the minor groove, where *Asn89* would be well-placed to interact with DNA. This observation aligns with the proposed function of the wing motif in providing increased specificity. In contrast, the wing motif in monomer B is less resolved and appears to exhibit a weaker interaction with the minor groove of the DNA oligonucleotide. Superimposition of the α 4-helices and N-terminal domain of each DNA-bound monomer of *NanR* gives an r.m.s.d. of 1.5, which demonstrates that the orientation of the DNA-binding domains is different (**Fig. 5e**). This structural change is facilitated by the α 4-linking helices. We hypothesize that in the presence of DNA, one monomer of *NanR* binds the operator sequence to partially stabilize the hetero-complex, while the opposing monomer undergoes a conformational change to untwist α 4-linking helices before engaging the DNA. This reorganization introduces an asymmetry in the DNA-binding pose of *NanR*, which we believe is a prerequisite to accommodate further *NanR* protomers, given the proximity we observe between each *NanR* within the multimeric hetero-complex in **Fig. 6**. The text on page 18-19, lines 438-467 now reads: Our cryo-EM structure of a *NanR* dimer bound to a (GGTATA)₂-repeat sequence shows that the repressor binds DNA in an asymmetric pose. Interestingly, the N-terminal DNA-binding domains engage DNA in a manner similar to *FadR*, a closely related *GntR* regulator. However, unlike *FadR*, where each N-terminal domain binds a palindromic DNA sequence symmetrically, *NanR* binds a repeat sequence with one N-terminal domain of the dimer engaging the consensus (GGTATA) sequence and the other N-terminal domain binding an adjacent non-consensus DNA sequence. The local resolution of each N-terminal domain was considerably different in the *NanR*-dimer₁/DNA hetero-complex structure, suggesting that the binding affinities for each N-terminal domain to the DNA is different, leading to an asymmetry of the binding pose. Notably, the putative DNA-binding residue *Arg73* can clearly be resolved in monomer A, but not in monomer B. Likewise, the density of the wing motif in monomer A is nestled within the minor groove, where *Asn89* would be well-placed to interact with DNA, an observation that aligns with the reported function of the wing motif to provide increased specificity²⁵. In contrast, the wing motif in monomer B is less resolved and appears to exhibit a weaker interaction with the minor DNA groove. Collectively, we hypothesize that in the presence of DNA, one monomer of *NanR* binds the operator sequence to partially stabilize the hetero-complex, while the opposing monomer undergoes a conformational change to untwist the α 4-helices before engaging the DNA. We believe the asymmetry in the DNA-binding pose is a prerequisite to accommodate further dimers of *NanR*, given the proximity we observe between each dimer within the *NanR*-dimer₃/DNA hetero-complex structure, as they span the entire (GGTATA)₃-repeat operator.

Minor points:

5. Overall, the figures 4, 5, 6 and 7 illustrate with clarity important features of the structure models. However, when it comes to the description of conformational changes between different states, some panels (e.g. Fig. 5d, Fig. 7a) are a bit messy and hard to follow. I would suggest the authors to provide additional movies of the overall structures and, if possible, a morphing between the different states described in Fig. 7.

Response: We agree these changes will help the reader by emphasizing the key conformational changes that occur in this system. To this end, we have supplied supplementary movies of the overall X-ray crystallography structure and effector-binding site (**Supplementary Movie 1**); the cryo-EM structure of the NanR-dimer₁/DNA hetero-complex, which includes a closeup on the protein-DNA interface (**Supplementary Movie 2**); a morph model to illustrate the conformational change between the DNA-free (X-ray structure) and the DNA-bound (cryo-EM structure) states described in **Fig. 6** (**Supplementary Movie 3**); and the cryo-EM structure of the NanR-dimer₃/DNA hetero-complex (**Supplementary Movie 4**). The movie legends are provided in the file ‘Description of Additional Supplementary Files’ (uploaded to the portal).

6. Figure 2e: the legend says 0.6 μM NanR33-263 although the figure shows 1.8 μM NanR33-263.

Response: We apologize for this oversight. We have corrected this to 1.8 μM in the **Fig. 2e** legend as the correct concentration is stated in the figure. The text on page 47, line 1132 now reads: **e** 1.8 μM NanR³³⁻²⁶³

7. Figure 4g: the panel shows that the DBD of the apo conformation has few interactions with the Neu5Ac-bound LDB, implicating a small number of residues. In contrast, the interface between the DBD of the Neu5Ac-bound conformation and the apo LBD, which is interpreted by the authors as “forming an extensive new interface locking the N-terminal domain in a closed conformation”, is not shown. It would be interesting to depict and compare side by side these two different interfaces.

Response: This is a good suggestion. We have now amended **Fig. 4g** to include a second half to the panel, which depicts the interface between the DBD of the Neu5Ac-bound conformation and the apo LBD. We agree that side by side these figures now highlight the differences in the domain interface between monomers.

8. Figure 5f-h: the transparent rendering of the EM map is very faint. It is hard to visualize even on the computer screen. The chosen panel views do not really help visualize potential interaction of amino acids with specific DNA-bases, as proposed by the authors.

Response: We thank the reviewer for pointing this out. We have now amended the transparent rendering of the cryo-EM density to enhance these features, which include several putative DNA-binding residues.

9. Also, it is not really clear from the text nor the figures if the binding mode to DNA of each NanR DBD is exactly the same (a perfect symmetry, like the FadR structure) or asymmetric. This part could be further improved together with the comment I made to the authors in point 4.

Response: We can confirm that the DNA binding mode of each N-terminal domain of NanR is asymmetric. This is evidenced by the difference in local resolution between α 3-helix of monomer A and B, which can be attributed to the degree of freedom between each N-terminal domain. We also observed that the wing motif is more ordered

in monomer A (higher signal-to-noise in the density map) than monomer B, which can also be explained by the inherent dynamics within the N-terminal DNA-binding domain. In contrast, the DNA binding mode observed in the crystal structure of FadR (**Fig. 5f**) has perfect symmetry, where both N-terminal domains (including the wing motif) have near identical B-factors. As initially discussed in point 4.2 above, the exact nature of the DNA binding mode in NanR, along with the possible function of this mode has now been clarified in the manuscript.

10. *Supplementary Fig 2: using PDB ID to annotate the homologs of NanR is not really informative. The authors could use the species and name of the proteins they identified as homologs and provide the PDB ID in the legend or in the Methods section.*

Response: We agree this would be more informative with additional species information and thus have changed the PDB IDs to species name and protein within the figure and moved the PDB IDs to the figure legend. The legend of **Supplementary Fig. 2** now reads: **Supplementary Fig. 2 | NanR has a longer N-terminal sequence than other GntR proteins.** Multiple sequence alignment was performed using Clustal Omega² and generated using ESPript 3.0³. The top five hits from a sequence homology search within the PDB were used in the sequence alignment: *S. agalactiae* GntR – PDB ID 6AZ6⁴; *E. coli* FadR – PDB ID 1E2X⁵; *C. glutamicum* LldR – PDB ID 2DI3⁶; *E. coli* McbR – PDB ID 4P9F⁷; *T. maritima* TM0439 – PDB ID 3FMS⁸. The red background highlights identical residues, whilst the yellow background indicates similar residues. Secondary structure elements depicted above the alignment were generated from the crystal structure of *E. coli* NanR (**Fig. 4**), solved in this study. The N-terminal DNA-binding domain (blue box), the linker region, and the C-terminal effector-binding domain (beige box) are highlighted.

11. *Supplementary table 8: 1) the camera should be Gatan “K2” and not “V2”. 2) The Ramachandran statistics do not correspond to the ones provided in the PDB deposition report.*

Response: We have corrected both these points within **Supplementary Table 8** of the manuscript. Again, we thank reviewer 1 for their careful reading of the manuscript.

12. *The authors provided a data processing pipeline for the cryo-EM study of the NanR-dimer1/DNA. It would be convenient for the reader to also provide one for the NanR-dimer3/DNA complex in the supplementary information. The authors should also explain why they stopped processing the data after the first 3D refinement of the NanR-dimer3/DNA complex in opposition to the NanR-dimer1/DNA complex for which they performed extensive 3D classification.*

Response: The reviewer makes a great suggestion to include a data processing pipeline for the NanR-dimer₃/DNA hetero-complex dataset. This has now been included as a new supplementary figure (**Supplementary Fig. 9**), while an additional supplementary figure (**Supplementary Fig. 10**) details the sample preparation and preferred orientation of the NanR-dimer₃/DNA hetero-complex dataset.

We apologize if our reasoning was not clear for stopping data processing after the first 3D refinement. We have taken the opportunity to further clarify this important point in the manuscript. The text on page 15, lines 369-371

now reads: However, due to the limited particle numbers and comparatively weaker signal of the resultant class averages compared to population 1, 3D reconstruction was not suitable for the population 2 dataset.

The text on page 35, lines 853-847 now reads: Further masked refinement strategies to tease out adjacent dimer-dimer interactions proved futile due to fewer number of particles in the pertinent class (**Supplementary Fig. 9c**). For population 2, the limited particle number as well as the comparatively weaker signal of the resultant class averages (**Supplementary Fig. 9d**) when compared to population 1, resulted in a 3D reconstruction that was not suitable for further processing and map interpretations (**Supplementary Fig. 9f**).

13. *There is a discrepancy between the legend of Supplementary Fig 4b referring to the concentration of NanR (120 μ M) used in the ITC experiment compared to the Methods section stating a NanR initial concentration of 172 μ M. Is 120 μ M after dialysis? Please, address this discrepancy.*

Response: We thank the reviewer for picking up on this error. 172 μ M was the final concentration loaded in our initial ITC experiment. We have now corrected this in the legend of **Supplementary Fig. 4b** to match the final concentration used in our new ITC experiment (416 μ M) to address the query from reviewer 1 in comment 1. The figure legend for **Supplementary Fig. 4** now reads: **b**, ITC isotherm of Neu5Ac (1 mM) titrated into NanR (416 μ M) in buffer C at 8 °C. The titration involved 25 injections of ligand solution (2 μ L) into the protein sample cell with a 200 sec interval between subsequent injections. When fitted, the data gave a K_D value of 16 μ M.

14. *Page 14, line 4: The content of Supplementary Fig. 5c does not explain the sentence from the main text.*

Response: We have removed the link to this figure as this no longer applies to the content of **Supplementary Fig. 5**. The text on page 10, lines 246-247 now reads: The asymmetry of the dimer is driven by the presence of Neu5Ac and Zn^{2+} in monomer A, but not in monomer B.

15. *Page 17, line 3: in the sentence “In contrast, when bound to DNA, the α 4-helices do not cross and, while the resolution does not permit a detailed examination of how these helices interact, we observed that the α 4-helix of monomer A (with Neu5Ac and Zn^{2+}) is longer and kinked compared with the α 4-helix of monomer B”, it is not clear whether the authors describe the atomic model from the X-ray structure or the model refined in the cryo-EM structure. They mention “ α 4-helix of monomer A (with Neu5Ac and Zn^{2+})” but the cryo-EM complex of NanR bound to DNA does not contain any Neu5Ac. Could you please clarify this paragraph?*

Response: We apologize that this was unclear in our first submission and thank the reviewer for seeking clarification. In this sentence, we are describing the conformational change of the DNA-bound cryo-EM model, as such we have removed the incorrect annotation of the Neu5Ac and Zn^{2+} in monomer A. We have also restructured this paragraph to form a new results section to explain the conformational changes that occur in the mechanism of NanR— *α 4-helices play a fundamental role in the allosteric mechanism of NanR*. The text on page 14-15, line 340-360 now reads: Together, our crystallography and cryo-EM experiments allow us to define the molecular choreography that occurs when NanR binds DNA to repress gene expression, or Neu5Ac to induce gene expression. The apo-NanR model, generated from the crystal structure, has a dimeric conformation, where the N-terminal DNA-binding domains are flexible (**Fig. 6a**, structure in blue), evidenced by the very few connections

between the N- and C-terminal domains (**Fig. 4f-g**). An overlay of the apo-NanR model with the DNA-bound cryo-EM structure revealed the most prominent change induced by DNA binding is the large reorganization of the N-terminal domains (**Fig. 6a**, structure in green) as they swing down to engage the major and minor grooves of the DNA—a conformational change that is facilitated by the α 4-linking helix (**Supplementary Movie 3**). In the crystal structure, the α 4-linking helices cross to form the domain-swapped monomers (**Fig. 6b**, upper panel; **Fig. 6c**). In contrast, when bound to DNA in the cryo-EM structure, the N-terminal domains are no longer domain-swapped through the α 4-linking helix (**Fig. 6b**, lower panel; **Fig. 6c**). This would require that the N-terminal domains untwist before or upon DNA binding, which is plausible given their flexibility evident in the apo-NanR structure. This conformational change of the α 4-linking helices can unambiguously be observed in the density maps between the crystal and cryo-EM structures (**Fig. 6b**). Neu5Ac binding promotes an opposite conformation (**Fig. 6a**, beige structure), whereby the N-terminal domain of one monomer moves close to the C-terminal domain of the opposing monomer allowing new interactions to be formed (**Fig. 4g**; **Fig. 6c**). This would lock the Neu5Ac-bound structure in a conformation that would be unfavorable for DNA binding. Taken together, these structural studies illustrate that the α 4-linking helix plays a fundamental role in the mechanism of NanR gene repression and allosteric induction.

16. Page 18, line 13: *To my understanding, the content of Supplementary Fig. 8j does not really explain the sentence from the main text. Maybe the authors wanted to refer to the panel 8h?*

Response: We have corrected this and now direct the readers to **Fig. 6b** and a new **Supplementary Fig. 9c**. These present both the density map for the NanR-dimer₃/DNA hetero-complex and the respective 2D class averages for this hetero-complex as part of a data processing workflow (originally the content of panel **Supplementary Fig. 8h**). The text on page 16, line 379-380 now reads: However, density for the central NanR dimer had a lower signal-to-noise ratio and was less resolved (**Fig. 6b**; **Supplementary Fig. 9c**).

17. Page 20, line 1: *the content of Supplementary Fig. 8e shows the angular distribution of the NanR-dimer1/DNA complex and is not related to the sentence from the main text. I would suggest the authors to double check the matching between the main text and their references to the supplementary information.*

Response: We apologize for this discrepancy. We have now corrected this by redirecting the reader to match the correct content in a new **Supplementary Fig. 10b**. The text on page 16, lines 390-393 now reads: Although the resolution of this dataset (**Supplementary Fig. 10b**) does not allow us to accurately locate the N-terminal extensions or define their role in the assembly process, we note that they would be well placed to form protein-protein interactions with the adjacent NanR dimers to stabilize the complex (**Fig. 6f**).

Reviewer 2

The NanR transcriptional repressor regulates genes associated with sialic acid metabolism in E. coli. This paper employs biophysical approaches to define the affinity, cooperativity, and stoichiometry of NanR binding to a (GGTATA)₃-repeat model DNA operator sequence and presents the structures of free NanR and NanR-DNA complexes. A mechanism is proposed for regulation of NanR-DNA interactions by the effector N-acetylneuraminate where the ligand allosterically disrupts DNA binding. This is an interesting system from the

perspective of allosteric regulation and authors have used an impressive array of methods to probe it. However, there are some problems with analysis and interpretation of the biophysical data that detract from the potential impact of this study.

1. *The most significant issue is that the model for allosteric regulation of NanR by acetylneuraminate in Fig. 8 is based on shaky biophysical data. The K_D for acetylneuraminate binding to NanR is reported as 183 μM . This seems far too weak to be biologically relevant. This K_D was obtained by ITC using a protein concentration of only 120 μM such that the C-factor (the product of the binding constant and the receptor concentration) is less than 1 when it should be at least 5 for reliable fitting. Thus, the reported confidence intervals of 91 to 275 μM seem too narrow. Equally troubling, binding of acetylneuraminate to NanR reduces its affinity for DNA by less than a factor of two. This change in affinity is tiny and seems incompatible with the structural change in NanR induced by acetylneuraminate that appears to render the protein incapable of binding DNA.*

Response: We thank the reviewer for prompting further examination by ITC. We have repeated our ITC experiments using a higher protein concentration (416 μM) and a lower Neu5Ac concentration (1 mM) to optimize the C-factor in line with the reviewer's query. When fitted to a 1:1 binding site model, the data gave a K_D value of 16 μM with a 95% confidence interval of 7 to 25 μM , provided a C-factor of ~ 32.5 and gave an N-value of 0.52 (data has been included in **Supplementary Fig. 4b**). Overall, we believe these data are a significant improvement from initial ITC data and addresses the reviewer's concerns, providing a K_D value that would be biologically relevant, a C-factor that is within the acceptable range for reliable fitting and an N-value, which is consistent with one Neu5Ac molecule bound per NanR. As such, this N-value is now supported by our SAXS experiments where we collected data of NanR in presence of Neu5Ac and found it best fit the theoretical scatter of the asymmetric crystal structure (**Supplementary Fig. 6a**), as opposed to a symmetrical Neu5Ac-bound dimeric model. Furthermore, with this new knowledge, we have now amended our schematic model (**Fig. 6c; Fig. 8**) to match what we observe in the crystal structure to explain the conformational change that occurs in NanR induced by Neu5Ac. The text on page 8, lines 194-197 now reads: We measured the dissociation constant (K_D) for Neu5Ac binding to NanR using isothermal titration calorimetry (**Supplementary Fig. 4b**), yielding a K_D of 16 μM (95% C.I. 7-25 μM) and an N-value of 0.52, which is consistent with one Neu5Ac bound per NanR dimer.

Accordingly, we have also updated the methods for the ITC experiment. The text on page 27-28, lines 663-673 now reads: Calorimetric titrations of NanR with Neu5Ac were performed with a Nano Isothermal Titration Calorimeter (TA Instruments). Purified NanR was initially concentrated to a final concentration of 416 μM via centrifugal ultrafiltration (30 kDa molecular weight cutoff; Sartorius) and then extensively dialyzed against buffer C. Neu5Ac was prepared in the same buffer by diluting a 100 mM stock solution to a final concentration of 1 mM. Protein sample (200 μL) was loaded in the sample cell, and 50 μL of Neu5Ac was loaded into the injection syringe. Titrations were initiated by a 1 μL injection, followed by 24 consecutive 2 μL injections every 200 s at 8 $^\circ\text{C}$ and a constant stirring speed of 60 rpm. A blank correction was obtained by injection of Neu5Ac (1 mM) into buffer C using an identical setup. Titration data were integrated using NITPIC^{61,62}, and analyzed in SEDPHAT by discarding the initial injection and fitting the binding isotherm 1:1 interaction model⁶³ to obtain K_D values.

2. There are several problems with the analysis of the gel shift and AUC data. It is impossible to assign any of the peaks in the $c(s)$ distributions in Fig. 2c to "complexes." This is a reversibly interacting system in rapid equilibrium so that features larger than the peak for the free DNA represent reaction boundaries, not discrete species. Also, it is very difficult to see many of the distributions in the plot due to the choice of colors and the thick linewidths.

Response: We agree with the reviewer: we had not meant to imply there are discrete complex species for the peaks present in the reaction boundary of Fig. 2c. These were assigned based upon our collective AUC experiments for this system. We note that the reaction boundary is a smooth transition between hetero-complex species, which cannot be individually distinguished due to the rapid reversibility of the system (i.e. the time scale of the sedimentation velocity experiment). As such, we hypothesize the reaction boundary observed in this study is due to fast kinetics of association/ on rates.

We thank the reviewer for highlighting the ambiguity in interpreting the sedimentation coefficient distribution plot shown in Fig. 2c. As such, we have now removed the arrows for each hetero-complex, adjusted the line weight in the distribution and offset the x-axis to better accommodate the interpretation between each NanR titration. Although, we note some of the lower concentrations (warmer colours) are very similar, so differentiating between these will always present some difficulty.

3. It is clear that NanR binds to the (GGTATA)₃ repeat DNA with positive cooperativity but Figure 2b throws out much of the interesting information present in the gel-shift (Fig. 2a) and sedimentation velocity (Fig. 2c) experiments to fit to an apparent K_d and a phenomenological Hill coefficient. The problem with this kind of analysis is exemplified by the fit shown in Fig. S1g which gives a Hill coefficient of 2.8 for binding of two NanR to (GGTATA)₂. This is impossible as the Hill coefficient must be less than or equal to the binding stoichiometry. All of the bound species are lumped together to determine "fraction bound." Instead, the data should be analyzed by fitting the individual band amplitudes in the gel-shift and the signal-average sedimentation coefficients for the sedimentation velocity data to a real model that incorporates cooperativity. Supplementary Table 2 contains "weight average" (really signal-average) sedimentation coefficient data correspond to Fig. 2C but they are not fit. Alternatively, the sedimentation velocity data could be directly fit to a model.

Response: We agree with the reviewer that, ideally, fitting each parameter to a model would be preferable, but in this case, with so many parameters, such modelling is not feasible. The approach of directly fitting our signal-average sedimentation coefficient data to a complete model that describes this complex system would require all of the parameters of the system to be fitted, which include: **1**) all concentrations of each of the species (minimum four); **2**) the sedimentation and diffusion coefficients for each species involved, including their respective anisotropies (i.e. free NanR monomer and dimer, free DNA, three hetero-complexes and any intermediate species); and **3**) the K_D 's and k_{off} rates for each reaction that occurs between the species in the system. Therefore, given the large number of parameters involved in this system, the proposed method by the reviewer is impractical and unlikely to yield a sensible answer. Had this been a simple protein/DNA interaction, such as one NanR dimer binding to the operator DNA sequence, then we could have modelled the system more completely. As our

reported K_D values from the gel shift assays and analytical ultracentrifugation data agree each other (within the same magnitude), we are confident in the nanomolar affinity we report for NanR.

We acknowledge that the Hill coefficient must be less than or equal to the binding stoichiometry. However, the protein/DNA interaction in this study is complex and involves a binding sequence that is non-palindromic. As such, the number of GGTATA repeats does not simply reflect the number of available binding sites, as only one monomer of NanR binds the GGTATA sequence and hence the N-terminal DNA-binding domains interact with non-equivalent DNA binding sites. Therefore, the reported Hill coefficient simply reinforces that we retain positive cooperativity in the system when only two GGTATA repeats are present (as value is >1), compared to the DNA operator sequence that has three GGTATA repeats as this is a complex system with a multimeric assembly process.

4. *There are several mistakes in the legend for Fig. 2. First, the Hill model is a mass-action binding model that incorporates cooperativity. The "mass action binding model" should be labeled as a noncooperative model. In Fig. 2e the concentration of NanR33-363 is labelled as 1.8 μ M but is indicated as 6.0 μ M in the caption. The statement in the caption "All plots are presented as $g(s)$ distributions against the molar concentration of monomeric NanR33-263" does not make sense. This same statement is found in the legend to Fig. 3.*

Response: We thank the reviewer for their careful reading of the manuscript. We have now corrected the "mass action binding model" to be correctly annotated as a non-cooperative binding model in both the methods and legend for **Fig. 2b**, and have changed the concentration of NanR³³⁻²⁶³ from 0.6 μ M to 1.8 μ M in the legend for **Fig. 2e** to match the correct concentration used in the analytical ultracentrifugation experiment (also shown in label within the figure). The "mass action binding model" has also been annotated as a non-cooperative binding model in the legend for **Supplementary Fig. 1g**. The text on page 24, line 569-572 now reads: **When the fraction containing bound DNA was plotted against NanR concentration, the data was best explained by the Hill model (shown below) with an Akaike information criterion (AIC) value of 99%, when compared to a non-cooperative binding model (AIC value of 1%).**

We apologize for the confusion regarding our explanation of the $g(s)$ distributions in **Fig. 2** and **Fig. 3**. Both $g(s)$ plots are plotted in molar units for both types of biopolymers/interaction partners (protein and DNA). We have now corrected this statement in the legend of **Figure 2** and **Figure 3** to read: **All plots are presented as $g(s)$ distributions with the molar concentration for each interacting partner (protein and DNA) plotted on the y-axis.**

5. *Why is the analysis of the NanR33-363 construct in Fig. 2d performed at 3 μ M DNA whereas the wild-type analysis was performed using fluorescence detection at 10 nM DNA? They should be done under the same conditions to make a valid comparison.*

Response: We apologize that we had not made the basis for these differences clear in our initial submission. The analysis performed with wild-type NanR using AUC with fluorescence detection (**Fig. 2c**) was a separate experiment, which was used to calculate the dissociation constant (K_D) in solution. This data were used to complement the K_D of wild-type NanR to the (GGTATA)₃-repeat DNA obtained by EMSA (**Fig. 2a**). In contrast, the analysis presented in **Fig. 2d** was conducted with both wild-type NanR and the NanR³³⁻²⁶³ construct using

AUC with absorbance detection and an identical titration range to evaluate whether the atypical N-terminal extension played a role in cooperativity. As this experiment was performed under the same conditions (absorbance detection and concentration range) we are confident in the differences that we observed between constructs with respect to hetero-complex formation.

6. *The integrations of the peak amplitudes for protein and DNA performed in figures 2e, 2f, 3b-d are based on the erroneous assumptions that the peaks correspond to species and they are baseline separated. Neither is true. The main peak in the distributions in Fig. 3e may correspond to a saturated species but that is not proven by the presence of free protein. The peak appears slightly shifted to the right relative to the peak in Fig. 3d. It would be necessary to go to a protein concentration where the peak stops shifting with concentration to verify that it represents a stable species, not a reaction boundary.*

Response: We apologize for any confusion here; we did not mean to imply that the signals for NanR and DNA from our multi-wavelength AUC experiments (**Fig. 2e-f** and **Fig. 3**) correspond to discrete, non-interacting species that are baseline separated and thus acknowledge that the NanR and DNA signals are indeed part of a reaction boundary. This observation highlights the reaction rates of the system, which indicates these reactions have fast kinetics of association (k_{on} rate), yet much slower kinetics of dissociation (k_{off} rate).

To address our assignment of the main peak in **Fig. 3e**, we believe it is reasonable to interpret the largest species observed (as was done in Zhang *et. al.* 2017-ref 55 in manuscript) based on molar ratio (obtained by integrating regions where the NanR and DNA signals co-migrate) and decide if this ratio would fit its hydrodynamics, since the molar ratio would not permit unrealistic models. As such, a 6:1 hetero-complex, could only have a molar ratio of 6:1, 12:2, 24:3, etc. Therefore, to verify this stoichiometry we calculated the molar mass of the hetero-complex by obtaining the sedimentation and diffusion coefficient values for this species in conjunction with a weight average v_{bar} (see page 34, line 755). A molar mass consistent with the theoretical molar mass of the 6:1 hetero-complex was obtained (reported in **Supplementary Table 5**) only when using a 6:1 weight average v_{bar} . This confirms that the largest species can only be a 6:1 hetero-complex (NanR-dimer₃/DNA hetero-complex), and all other peaks observed in the experiment are likely accumulations of intermediate species (e.g. 2:1, 4:1, etc). Analogous to the interpretation of the largest species, the sedimentation and diffusion coefficient values for these intermediate species, as well as the molar ratios from integrating regions of co-migration (reported in **Supplementary Table 5**), can be used to unambiguously identify the intermediate stoichiometries. We note that we could not obtain a molar mass value for the 4:1 hetero-complex (only the 2:1 hetero-complex) as an accurate diffusion coefficient could not be obtained. However, this is not surprising, given that this species lies within a reaction boundary, which indicates that the reaction is simply too fast to produce enough of this species to be detected. Nevertheless, we feel our assignment of the peak amplitudes in both **Fig. 2e-f** and **Fig. 3** is reasonable, as the molar ratios match the stoichiometries listed and are largely supported by a molar mass consistent with that stoichiometry.

7. *The description of the AUC methods is excessively long and Fig. S3 is unnecessary.*

Response: We take the reviewer's point, but feel that because we have employed various AUC collection strategies in this project, the level of detail in the methods is appropriate, and more importantly, is needed for

others to repeat the work should they wish to. Multi-wavelength detection is an emerging AUC strategy, as such the analysis workflow presented in **Supplementary Fig. 3** is warranted to better guide the reader. We are happy to take direction from the Editor on whether this is considered superfluous; we erred on the side of completeness.

Reviewers' Comments:

Reviewer #1:

Remarks to the Author:

The revised version of the manuscript by Horne, Venugopal and colleagues has addressed all the concerns I raised in the previous review. The amended discussion on the stoichiometry and the revised structural analysis, together with the newly provided movies and modified figures, have greatly improved the quality and readability of the manuscript, which should now be suitable for publication.

Reviewer #2:

Remarks to the Author:

In this revised manuscript the authors have addressed several of the concerns raised in the reviews of the original manuscript but a major issue remains. The authors did not respond to the concern that acetylneuraminate (Neu5Ac) binding decreases the DNA binding affinity of NanR by less than two-fold. This small change is incompatible with a classical allosteric switching hypothesis where effector binding induces the repressor to dissociate from the DNA operator. Instead, a model is proposed where Neu5Ac weakens the NanR-DNA interaction, such that DNA adenine methyltransferase (DAM) can bind and methylate the operator site and subsequently block NanR from re-binding, thereby inducing gene expression. This is essentially a kinetic model that postulates that in the free state NanR is locked onto the DNA with a very low off-rate but in the Neu5Ac-bound state the NanR off-rate increases dramatically so that DAM gains access to the operator. However, there is no data to support such a picture. The equilibrium data imply that the kinetics of NanR-DNA interactions are likely to be very similar in the presence and absence of bound Neu5Ac such that DAM would have access to the operator regardless of the binding of Neu5Ac.

The control of DNA binding by Neu5Ac is clearly an important point for this study as this observation is referenced in the abstract, in the introduction on page 4, and in the discussion on page 18, and is a critical component of the model presented in Figure 8. This issue must be resolved prior to publication. A potential explanation for the small increase in K_d upon Neu5Ac binding is that the ligand is only weakly bound ($K_d = 16 \mu\text{M}$) and it dissociates from NanR during the electrophoretic separation used to monitor DNA binding. Thus, the K_d for the Neu5Ac-NanR complex binding to DNA is underestimated. A better approach would be to use a true equilibrium technique such as AUC or ITC to monitor the binding of NanR to DNA in the presence of saturating Neu5Ac.

We thank the reviewers for their positive and constructive comments, which we have incorporated into our newly revised manuscript entitled “**Mechanism of NanR gene repression and allosteric induction of bacterial sialic acid metabolism**” by Horne, Venugopal *et. al.*

We have addressed each reviewer comment (shown in *black italics*) in our point-by-point response below, where we respond in **blue text**, with changes **highlighted in grey** in this letter and the main text document. In addition, we have made minor editorial changes, which are also **highlighted in grey**.

Reviewer 1

The revised version of the manuscript by Horne, Venugopal and colleagues has addressed all the concerns I raised in the previous review. The amended discussion on the stoichiometry and the revised structural analysis, together with the newly provided movies and modified figures, have greatly improved the quality and readability of the manuscript, which should now be suitable for publication.

Response: **Thanks again to reviewer 1 for your constructive suggestions that enabled us to improve the quality and readability of the manuscript.**

Reviewer 2

1. The authors did not respond to the concern that acetylneuraminate (Neu5Ac) binding decreases the DNA binding affinity of NanR by less than two-fold. This small change is incompatible with a classical allosteric switching hypothesis where effector binding induces the repressor to dissociate from the DNA operator. Instead, a model is proposed where Neu5Ac weakens the NanR-DNA interaction, such that DNA adenine methyltransferase (DAM) can bind and methylate the operator site and subsequently block NanR from re-binding, thereby inducing gene expression. This is essentially a kinetic model that postulates that in the free state NanR is locked onto the DNA with a very low off-rate but in the Neu5Ac-bound state the NanR off-rate increases dramatically so that DAM gains access to the operator. However, there is no data to support such a picture. The equilibrium data imply that the kinetics of NanR-DNA interactions are likely to be very similar in the presence and absence of bound Neu5Ac such that DAM would have access to the operator regardless of the binding of Neu5Ac. The control of DNA binding by Neu5Ac is clearly an important point for this study as this observation is referenced in the abstract, in the introduction on page 4, and in the discussion on page 18, and is a critical component of the model presented in Figure 8. This issue must be resolved prior to publication. A potential explanation for the small increase in K_d upon Neu5Ac binding is that the ligand is only weakly bound ($K_d = 16 \mu\text{M}$) and it dissociates from NanR during the electrophoretic separation used to monitor DNA binding. Thus, the K_d for the Neu5Ac-NanR complex binding to DNA is underestimated. A better approach would be to use a true equilibrium technique such as AUC or ITC to monitor the binding of NanR to DNA in the presence of saturating Neu5Ac.

Response: We agree with the reviewer that the two-fold reduction in DNA binding by Neu5Ac is incompatible with a classic allosteric switching model, in which we might expect Neu5Ac to abolish DNA binding.

Our solution (ITC) and structural studies (crystallography), along with the *in vivo* work of others (Kalivoda *et al.* references 21 and 22 in the manuscript), support the claim that Neu5Ac binds to NanR and is the principle effector that alters gene expression. Further, our *in vitro* EMSA experiments demonstrate that Neu5Ac disrupts the protein-DNA interaction, albeit by only a two-fold change. As requested by the reviewer and in support of our EMSA experiment, we present analytical ultracentrifugation data in **Supplementary Figure 4f** that compares the NanR-DNA interaction in the absence and presence of saturating Neu5Ac (20 mM). This demonstrates that the addition of Neu5Ac only weakens the interaction, since the sedimentation coefficient for the complex decreases and there is an increase in the signal for free DNA. We have amended the text on page 9, lines 212-219 now reads: This change in DNA binding affinity was also evident in solution. Analytical ultracentrifugation data (**Supplementary Fig. 4f; Supplementary Table 6**) compares the NanR-DNA interaction in the absence and presence of saturating Neu5Ac (20 mM). In the absence of Neu5Ac (purple and red traces), the DNA is saturated or near saturated with bound NanR. The addition of Neu5Ac (green and orange traces), however, weakens the NanR-DNA interaction, evidenced by a decrease in the sedimentation coefficient for the NanR-DNA complex (orange trace) and an increase in the signal for free DNA (green trace).

But, as the reviewer correctly notes, this data does not directly support the previously proposed model whereby DAM methylates the site in the absence of NanR. As such, we have adjusted our proposed model such that we

suggest the participation of additional *in vivo* elements to facilitate induction of the sialic acid metabolic machinery in *E. coli*. We have amended both our discussion on page 18 and **Figure 8** to present a model mechanism for the regulation of gene expression that is based on the data from this study. The text on page 18, lines 432-447 now reads: In agreement with this model, our binding studies demonstrate that NanR binds Neu5Ac with micromolar affinity with a stoichiometry of one Neu5Ac molecule per NanR dimer, which is consistent with our crystallographic and solution studies. However, in contrast with the Kalivoda *et al.* model, our biophysical studies reveal that NanR retains its dimeric structure with or without Neu5Ac present. We also observe that the presence of Neu5Ac does not abolish DNA binding, but rather attenuates DNA binding two-fold. As such, this small change in affinity is inconsistent with the classic allosteric mechanism employed by members of the GntR family, whereby effector binding induces the repressor to dissociate completely from the DNA operator²³. This suggests that the mechanism of induction for NanR may involve additional elements that have yet to be determined. From the literature, one hypothesis is an interplay between the cAMP receptor protein (CRP), which could hinder the binding of NanR to its operator by recruiting RNA polymerase to the promoter region of DNA through protein-protein interaction^{21,22,42}. An alternative hypothesis is the methylation of a GATC sequence located five base-pairs upstream of all three *E. coli* NanR-regulated operons by DNA adenine methyltransferase (DAM) to act as roadblock for NanR⁴³. Consistent with this hypothesis, Oshima *et al.* report a decrease in the expression of sialic acid metabolic genes in a DAM mutant⁴⁴.

Reviewers' Comments:

Reviewer #2:

Remarks to the Author:

In the second revision of the manuscript the authors addressed the concern I raised regarding the weak effect of Neu5Ac binding on the affinity of the NanR-DNA interaction by performing an additional experiment and revising the text and Figure 8. The revised paper mostly removes the overinterpretation of the small affinity change induced by the effector. However, the new models proposed to explain how the effector induces NanR expression lack clarity. As detailed below, the absence of a functional interpretation of the structural and biophysical data in this study reduces its significance.

Unfortunately, the new experiment does not add much to the paper because it was performed at high concentrations of NanR and DNA that are well above the K_d for the protein-DNA interaction. The result is a very small change in the extent of complex formation induced by the ligand as shown in supplementary Figure 4f and Table 6. This experiment does not address the question of whether the two-fold effect of Neu5Ac on the NanR-DNA interaction K_d measured by EMSA is an underestimate due to dissociation of the ligand during electrophoresis.

By accepting the EMSA-derived result rather than testing whether it is true, the authors are forced to discard the classical allosteric switching model and instead propose that induction of NanR involves additional mechanisms involving cAMP receptor protein or DAM methylation. These models are not clearly spelled out and they do not invoke the structural or biophysical data reported in the paper. As a result, the model for effector modulation of NanR depicted in Figure 8b is vague. It retains the idea that Neu5Ac may induce dissociation of NanR from DNA but adds the possibility that the mechanism may involve elements that have not yet been determined.

In their rebuttal the authors state that their data does not directly support the previously proposed model whereby DAM methylates the site in the absence of NanR. Why is the DAM model being resurrected in the new text on page 18?

We thank Reviewer 2 for their constructive comments, which we have incorporated into our newly revised manuscript entitled “**Mechanism of NanR gene repression and allosteric induction of bacterial sialic acid metabolism**” by Horne, Venugopal *et. al.* In particular, despite the shutdowns in Melbourne, Australia last year caused by the Covid-19 pandemic, we have been able to conduct the biophysical experiments requested by Reviewer 2 (see point 1 below). This has significantly improved the work.

We have addressed each reviewer comment (shown in *black italics*) in our point-by-point response below, where we respond in **blue text**, with changes **highlighted in grey** in this letter and the main text document. In addition, we have made minor editorial changes, which are also **highlighted in grey**.

Reviewer 2

1. In the second revision of the manuscript the authors addressed the concern I raised regarding the weak effect of Neu5Ac binding on the affinity of the NanR-DNA interaction by performing an additional experiment and revising the text and Figure 8. The revised paper mostly removes the overinterpretation of the small affinity change induced by the effector. However, the new models proposed to explain how the effector induces NanR expression lack clarity. As detailed below, the absence of a functional interpretation of the structural and biophysical data in this study reduces its significance.

Response: We thank the reviewer for promoting further discussion of this important point. In our view (and that of Reviewer 1) there is significant “*functional interpretation*” in defining the structural basis by which NanR engages DNA and represses gene regulation. However, and in agreement with Reviewer 2, we were cautious on the “*functional interpretation*” of the mechanism by which Neu5Ac relieves repression, since we lacked at the time the key experiment in our previous submission, as stated by Reviewer 2.

Having now conducted the additional experiments requested by the Reviewer 2 (detailed below) we have added the necessary interpretation to provide formal validation for the mechanism of induction of NanR, in the presence of Neu5Ac, which we present schematically in **Figure 8** and discuss in the manuscript. Thus, we can now provide a full “*functional interpretation of the structure and biophysical data*”, increasing its significance.

Our amended discussion of the mechanism of induction on page 14, lines 432-439 now reads: **By measuring the protein-DNA interaction, using nanomolar concentrations of NanR above and below the reported K_D , and saturating concentrations of Neu5Ac, we observed that the presence of Neu5Ac attenuates DNA binding 28-fold. This large change in DNA-binding affinity, in concert with the Neu5Ac-induced conformational change identified in our crystal structure, demonstrates the mechanism of induction adopted by NanR is consistent with the classic allosteric mechanism employed by other members of the GntR family. These data support a model in which effector binding induces dissociation of the repressor from the DNA operator.**

2. Unfortunately, the new [EMSA] experiment does not add much to the paper because it was performed at high concentrations of NanR and DNA that are well above the K_d for the protein-DNA interaction. The result is a

very small change in the extent of complex formation induced by the ligand as shown in supplementary Figure 4f and Table 6. This experiment does not address the question of whether the two-fold effect of Neu5Ac on the NanR-DNA interaction K_d measured by EMSA is an underestimate due to dissociation of the ligand during electrophoresis.

By accepting the EMSA-derived result rather than testing whether it is true, the authors are forced to discard the classical allosteric switching model and instead propose that induction of NanR involves additional mechanisms involving cAMP receptor protein or DAM methylation. These models are not clearly spelled out and they do not invoke the structural or biophysical data reported in the paper. As a result, the model for effector modulation of NanR depicted in Figure 8b is vague. It retains the idea that Neu5Ac may induce dissociation of NanR from DNA but adds the possibility that the mechanism may involve elements that have not yet been determined.

Response: We take the reviewer's point and, with respite in our COVID-19 lockdown, were able to perform a more quantitative experiment at nM concentrations of NanR to address this concern more directly. These fluorescence-detection AUC experiments demonstrate that the presence of the allosteric modulator, Neu5Ac, attenuates the NanR-DNA interaction 28-fold in solution, relative to the Neu5Ac-free kinetics of NanR-DNA interaction (**Supplementary Fig. 4d-e**). As this new data provides evidence to formally validate the conformational changes observed in our structural data and depicted schematically in **Figure 8**, we have since removed the EMSA-derived result from **Supplementary Fig. 4**. We thank the Reviewer for suggesting this key experiment to understand the mechanism of induction of NanR in the presence of Neu5Ac.

We have included additional text to report these new observations on pages 7-8, lines 207-219 now reads: We next examined the effect of Neu5Ac binding on the NanR-DNA interaction by titrating NanR against the FAM-5'-labeled (GGTATA)₃-repeat operator sequence, in buffer supplemented with excess Neu5Ac (20 mM), and monitoring binding by fluorescence-detection analytical ultracentrifugation, using an analogous setup as the experiment in the absence of Neu5Ac (**Fig. 2c**). In comparison to the data without Neu5Ac, there was a notable difference in the sedimentation coefficient distribution for the titration series when Neu5Ac was present in solution (**Supplementary Fig. 4d; Supplementary Table 6**), evidenced by an overall decrease in the signal for Nan-DNA hetero-complex formation (3.5–12 S) and an increase in the signal for free DNA (2-3.5 S), suggesting Neu5Ac attenuates the NanR-DNA interaction. This attenuation was further illustrated in the binding isotherm, where the binding affinity for DNA decreased approximately 28-fold in the presence of Neu5Ac (**Supplementary Fig. 4e**, red line, $K_D=578\pm 26$ nM and $n=2.0\pm 0.6$), relative to the assay without Neu5Ac (**Fig. 2b**, blue line; **Supplementary Fig. 4e**, black line, $K_D=20\pm 1$ nM and $n=1.9\pm 0.2$).

3. *In their rebuttal the authors state that their data does not directly support the previously proposed model whereby DAM methylates the site in the absence of NanR. Why is the DAM model being resurrected in the new text on page 18?*

Response: In light of our new data demonstrating a clear and significant allosteric effect of Neu5Ac on DNA binding, which is compatible with a classic allosteric switching model (as noted by the reviewer), we have removed the discussion of the DAM model within the manuscript.

We have included additional text to summarize our findings page 15, lines 462-471 now reads: Collectively, our findings offer formal support for a mechanism of sialoregulon repression in *E. coli* (**Fig. 8**) that is unique among reported GntR-type regulator mechanisms. The combination of cooperative binding to a repeat DNA sequence, a process mediated by atypical N-terminal extensions of the DNA-binding domain and the formation of a multimeric protein-DNA hetero-complex distinguish NanR from other reported modes of transcriptional regulation amongst the GntR superfamily. Importantly, we also functionally validate Neu5Ac as the allosteric modulator of NanR, which had previously been proposed, but lacked formal supporting evidence at the molecular level. By defining the mechanisms of induction and of gene repression for NanR, our studies extend our knowledge of the GntR superfamily and our understanding of the complex interactions between protein and DNA that lie at the heart of many biological processes.

Reviewers' Comments:

Reviewer #2:

Remarks to the Author:

The authors have addressed the concerns raised I raised in the last review and the paper is now suitable for publication.